# Face and context integration in emotion inference is limited and variable across categories and individuals

Srishti Goel [1] ✉, Julian Jara-Ettinger [1,2], Desmond C. Ong [3] & Maria Gendron [1] ✉

The ability to make nuanced inferences about other people's emotional states is central to social functioning. While emotion inferences can be sensitive to both facial movements and the situational context that they occur in, relatively little is understood about when these two sources of information are integrated across emotion categories and individuals. In a series of studies, we use one archival and five empirical datasets to demonstrate that people could be integrating, but that emotion inferences are just as well (and sometimes better) captured by knowledge of the situation alone, while isolated facial cues are insufficient. Further, people integrate facial cues more for categories for which they most frequently encounter facial expressions in everyday life (e.g., *happiness*). People are also moderately stable over time in their reliance on situational cues and integration of cues and those who reliably utilize situation cues more also have better situated emotion knowledge. These findings underscore the importance of studying variability in reliance on and integration of cues.

People's perceptions, decisions, and actions often hinge on understanding of other's minds, including their emotional experiences. For example, people make moral and legal judgments based on inferences of remorse or guilt in the perpetrator[1–3] and provide support to people perceived as distressed or grieving[4–6]. In negotiations, people concede more to counterparts perceived as angry[7,8]. Outcomes that follow emotional inferences have implications for a wide variety of social contexts ranging from intimate relationships to courtrooms and from classrooms to global diplomacy. Further, a booming industry, projected to grow to $37.1 billion in the next few years[9], is organized around building intelligent machines that can use non-verbal behavior to "read" people's inner states, such as their emotions, to predict their behavior[10]. And many definitions of what distinguishes an intelligent from unintelligent machine is emotional capacity in addition to cognitive performance[11], particularly when the goal is to achieve human-like performance[12]. These applications of emotion science in everyday life and industry suggest it is crucial to build a systematic and robust account of how humans infer the emotions of others.

Despite the complexity of emotional events in the real world, research examining emotion inferences has disproportionately focused on how people process isolated canonical facial portrayals of emotion (or, facial expressions)[13]. This approach, termed the *common view* is prevalent in basic science, clinical science, education, the tech sector, security, and in popular media and entertainment (for review see ref. 14). Specifically, both in theory and in practice, researchers often assume that emotion categories and facial behaviors demonstrate stable, context-free links. For example, this *common view* assumes that there is an expression of anger (i.e., scowling) that is invariant across the specific contexts in which these expressions might occur (e.g., in both a conflict with a romantic partner and when getting cut off in traffic). Indeed, the field of "emotion recognition" is premised on this assumption, where decontextualized canonical portrayals of

[1]Department of Psychology, Yale University, 100 College St, New Haven, CT, USA. [2]Wu Tsai Institute, Yale University, 100 College St, New Haven, CT, USA. [3]Department of Psychology, The University of Texas at Austin, 108 E Dean Keeton St, Austin, TX, USA. ✉e-mail: srishti.goel@yale.edu; maria.gendron@yale.edu

emotion are presented to perceivers and "accuracy" is computed based on whether responses match the a priori category. Decades of evidence suggests that this *common view* focused on the face is insufficient. Emotion inference is more complex than simply relying on a limited set of cues from the face[15–17]. Evidence suggests that perceivers rely heavily on additional contextual information such as bodily movements[18], vocalizations[19], culturally learned knowledge[20,21], and knowledge of the social situation, or situational context[22–24] to infer emotions (for review, see refs. [15,25]).

Though prior studies demonstrate the crucial role of context in informing emotion inferences, most are limited because they have classically focused on dominance of certain cues rather than integration across cues to form inferences (for a classic example on integration, see ref. [26]). That is, earlier work was often aimed at establishing dominance of cues (facial or situational) by pitting them against one another as highly contrastive information (for example, see refs. [23,27]). In this approach, there was frequent experimental use of pre-selected "clear" (meaning, eliciting high consensus in perceiver judgments) facial and situational cues (for critique of this clarity of stimuli approach, see ref. [28]) which likely restricted the stimuli to more stereotypical displays (e.g., a wide-eyed, gasping expression for fear). Moreover, this contrastive approach (e.g., combining a "fear" face with an "angering" situation) may have unintentionally encouraged participants to ignore one piece of information over the other rather than integrating information together. That is, when information is "contrasting", it may be that only one interpretation based on a single cue is assumed correct, rendering judgments an artifact of this experimental setup. Thus, the contrastive approach might limit relevance to real world emotion inference. The present research is designed to address integration of cues in emotion inferences using a computational modeling approach that compares a cue-integration model to simpler models depending on facial information or situational information alone.

To study how perceivers attend to and combine facial information and situational context, it is necessary to model how people understand the relationship between situations and non-verbal behaviors. That is, perceivers may not simply rely on facial and contextual information equally. Instead, perceivers may integrate situational and non-verbal cues based on their lay beliefs about the causal relationship between the situation, the emotional experience of the target and, the non-verbal behavior that they display[29,30]. Some recent work has advanced our ability to model this type of lay theory-informed integration. Specifically, Ong and colleagues[31] proposed a rational-observer model of how laypeople integrate multiple emotional cues, based upon other rational-observer models in human visual perception. This model assumes that observers hold a causal lay theory in which situational outcomes cause emotions, which in turn cause facial expressions (described in more detail below). This model extends beyond assumptions of the *common view*. The *common view* would suggest that facial expressions provide diagnostic information about the emotion that caused them. When situational outcomes are available, these should only provide convergent, overlapping information about the underlying emotion to that derived from the face. As such, an integration model and model based only on the face should capture emotion judgments to the same degree. In contrast, if the cue-integration model captures judgments to a greater degree than the face-only model, this implies that facial cues provide incomplete information about emotion, and that the situational context is providing additional information that perceivers utilize to infer emotion. Given that this lay theory is, essentially, a Directed Acyclic Graph, it should be noted that situations do not directly impact the form of the expression itself: instead, situations affect expressions only through the emotion itself. The cue-integration model (Eq. [1] below) derived from this DAG thus assumes perceivers draw on context-independent mental representations of emotional expressions in the face, in line

with an aspect of the *common view* outlined earlier[31]. Critically, in Ong and colleagues' original work, across several experiments, the Bayesian cue-integration model closely tracked perceivers' judgments, suggesting that people can and do integrate information from the situation and face, in line with this theory. Here, we build on these modeling advances to ask two primary questions about the robustness of this model to capture emotion inferences.

First, the limited repertoire of facial movements focused on in the prior literature likely fails to capture the complexity of facial movements during real world instances of emotion. Facial movements during instances of emotion rarely conform to the canonical expressions commonly used in scientific studies[14,32]. Prior studies indicate that when facial cues are ambiguous (as often in the real world[23,33]), highly variable[24], neutral[34,35], or even entirely missing[36], people can still make robust inferences about emotions based on the context. This implies that this Bayesian integration model may not be as applicable to emotion inferences when the stimuli have greater diversity and complexity. Thus, we aim to test the robustness of the Bayesian integration account here.

We ask whether the Bayesian integration of facial and situational cues (based on the model from ref. [31]) best accounts for perceiver's emotion inferences compared to inferences based on single cues (face or context alone) when using stimuli with more variability and complexity. Prior modeling work was tested using a narrow set of situations: different outcomes (amount of money won) within a gambling game or outcomes (win or loss) from a game-show[37] or tennis match[38], sometimes along with computer-generated caricatured facial portrayals[31]. Thus, prior work leaves open the question of whether integration of facial and situational cues, based on the Bayesian DAG model, extends to more naturalistically varying stimuli that better reflect the diversity of experiences and expressive behaviors in everyday life. In this work, we test whether the existing model of cue-integration[31] (heretofore referred to as integration) accounts for emotion inferences in a dataset of over 2500 perceiver judgments, where the situations are high in emotional complexity and the expressive behavior is highly variable, yet high in perceived intensity[24]. Based on evidence that canonical expressions are rarely documented in spontaneously produced expressions[32] and previous research demonstrating that high-quality acted portrayals of emotion rarely involve canonical expressions[39], we contend that the present dataset provides a relatively more ecologically valid test of how perceivers integrate face and situations.

Given the complexity of this dataset, we are also able to examine cue-integration for both overall emotion judgments as well as for specific emotion categories (e.g., anger, fear, etc.). It is possible that individuals are more likely to engage in integration when they have greater certainty about the links between given cues and emotions. This knowledge could further be undergirded by the frequency of encountering specific cues in everyday life. For instance, people vary in how often they reported seeing in daily life the "canonical" facial expressions like the wide-eyed, gasping expression often associated with fear, and the furrowed eyebrows and tightened lips often associated with anger[40]. Moreover, recent meta-analyses have also suggested that these canonical expressions are only weakly related to actual emotional experiences[32]. For emotion expressions that have low reliability or are less frequently encountered, perceivers should be less certain about how these expressions link to emotions. As a result, they may be more likely to rely on situational cues to form inferences rather than relying on or integrating facial cues.

Second, in addition to examining integration at the group level, we extend our work on integration to examine individual differences. Here we ask whether individuals vary in the extent to which they integrate versus rely on situational or facial cues alone, and whether these differences are stable across time. There is reason to suspect that the utilization of situational cues may be subject to individual

differences. People vary in their ability to represent the meaning of complex situations. Not only do people vary in their ability to infer the emotional consequences of situations in line with group consensus[41] but they also vary in behavioral response selection stemming from these variable representations[42]. Variation in the representation of situations is also observed when comparing healthy adults to individuals from clinical samples (for example, see refs. 43,44). For instance, more hostile individuals are more likely to interpret ambiguous situations as threatening[44] and people with greater state anxiety represent stressful situations intensely on multiple dimensions such as, physical danger, social evaluation, and conflict[43]. Together, findings indicate that people's representations of social situations and the meaning they draw from it can vary substantially. We build on these findings by focusing on individual differences in how the situational context is relied upon in the formation of emotional inferences.

We know relatively little about the patterns of cue reliance across individuals, despite this being a potentially consequential individual difference. It is worth investigating whether a single model quantifying emotion inference is sufficient or if there is meaningful variability across individuals. To this end, we examined whether individual differences in cue-integration relate to existing measures of individual differences in emotion domain. Specifically, we looked at people's ability to accurately infer (based on group consensus) the emotions that are most likely experienced in various social situations (Situation Test of Emotional Understanding or STEU[45]) and their ability to detect contextual cues (Context Sensitivity Index[46]). The emotion literature has largely neglected differences in how individuals integrate situational context in emotion inferences. Instead, the study of individual differences in emotion inference (e.g., within emotion intelligence literature) is often focused on inferences formed from a single modality such as the face (for example, see ref. 47) or the situational context (for example, STEU[45]). Even when multimodal cues are presented to participants (for example, Geneva Emotion Recognition Test[48]), researchers have only focused on the individual differences in the degree of contextual modulation of emotion inferences for a limited set of contrasting facial and contextual cues[49]. It remains an open question whether there is variation in individuals' tendency to integrate complementary facial and situational cues when inferring emotions and the degree of cue reliance.

We examine these questions about integration in emotion inference across five studies that use archival[24] and empirically collected datasets ($N_{total} = 752$). The archival data[24] contains ratings of social scenarios, ratings of actors' portrayals of those scenarios, and ratings of the combination for a set of 13 emotion categories—*amusement, anger, awe, contempt, disgust, embarrassment, fear, happiness, interest, pride, sadness, shame, surprise*. These ratings were obtained from a total of 604 stimuli pairs (scenarios and poses) that are sourced from two books: *In Character: Actors Acting*[50] and *Caught in the Act: Actors Acting*[51]. These volumes contain images of expressions posed by a pool of professional actors after they were provided with emotionally evocative scenarios. To calculate model predictions, it is necessary to compute people's prior probabilities over emotions. Our first experimental data consists of ratings for people's ($N = 44$) likelihood of perceiving each of the 13 emotion categories in their everyday lives (*priors task*). The second empirical dataset (Study 2) consists of ratings ($N = 142$) of the combination of facial portrayals and social scenarios for a randomly selected subset of stimuli ($n = 44$), an ability-based measure of individuals' sensitivity to context (CSI scale[46]), and their self-reported ability to differentiate between emotion states (RDEES[52]). The third empirical dataset (Study 3) is an extension of Study 2 and consists of ratings ($N = 162$) of the combination of facial portrayals and social scenarios for the same subset of stimuli used earlier, an ability-based measure of individuals' sensitivity to context (CSI scale[46]), and their ability to understand the relationship between emotions and situations (STEU-B[41,45]). The fourth empirical dataset (Study 4) is a pre-registered study to replicate and extend the findings of Study 3 in a nationally representative sample ($N = 294$). The final empirical dataset (Study 5) was collected to evaluate test-retest reliability of individual difference in model estimates to examine how trait-like they are. The study consists of participants ($N = 110$) ratings for the same pairs ($n = 44$) of facial portrayals and situational descriptions as used in Studies 2–4. Ratings are provided for the 13 emotion categories twice—in Session 1 and Session 2 that are administered two weeks apart.

## Results

### Model overview

The application of Bayesian computational models in understanding how people reason about emotions (or, affective cognition; see refs. 31,38) is relatively recent compared to how people reason about other mental states such as beliefs, desires, and intentions[53–55]. These normative models help formalize and test human lay theories by capturing different hypotheses about the computations in people's mind and comparing these against human judgments (approach of rational analysis[56]).

Researchers recently used this approach to capture a *lay theory of emotion* that they propose humans use to understand others' emotions[31]. The theory captures expectations of how situational outcomes cause emotions, and how emotions then cause expressive behaviors such as facial portrayals. Equipped with this causal model, people can then use Bayesian inference to determine what emotions people are likely experiencing, given information about their facial expressions and the situation that they're in. Ong and colleagues (2015) presented the formal computational model of this idea and extended it to derive a model equation that captures inferences about emotions following integration of facial and situational cues (or, cue integration), given by:

$$P(e|s,f) \propto \frac{P(e|s)P(e|f)}{P(e)} \qquad (1)$$

Here, $P(e|s,f)$, represents observers' beliefs that an agent is experiencing emotion $e$, given their facial expression $f$ and situation outcome $s$. If people infer each other's emotions by applying Bayesian inference to the *lay theory of emotion*, then people's inferences should be proportional to the product of likelihood of each individual cue $P(e|s)$ (the probability of emotion $e$ given situation $s$) and $P(e|f)$ (the probability of emotion $e$ given facial expression $f$), divided by the prior probability of an emotion occurring $P(e)$. See Ong and colleagues (2015) for the full derivation of this equation. We compute overall model probabilities by averaging emotion intensity ratings across participants. This assumes that the distribution of frequency ratings can be used to compute model probabilities, consistent with prior literature demonstrating people's inferences reflect probability matching behavior[57–62] and that people act as Bayesian samplers[63–65].

Ong and colleagues compared the Bayesian cue-integration model with two simpler alternative models that capture beliefs about reliance on single cues, i.e., the cue dominance models, instead of integration. These represent beliefs that an agent experiences emotions due to situation outcomes alone (Situation-only model or $P(e|s)$), and beliefs that an agent's experience of emotions can be inferred solely from external cues like facial movements (Face-only model, or $P(e|f)$). These simpler models suggest that an observer would rely on a single cue (face or situation) to approximate their inference of emotions when presented with both faces and situations.

In this approach, the model estimates are compared to people's judgments of emotions when viewing both facial and situational cues to examine which model overall best captures people's inferences (Fig. 1a). Here, not only do we examine the best fitting model overall, but we extend this model comparison at the level of an individual by comparing an individual's judgments of emotions when viewing both

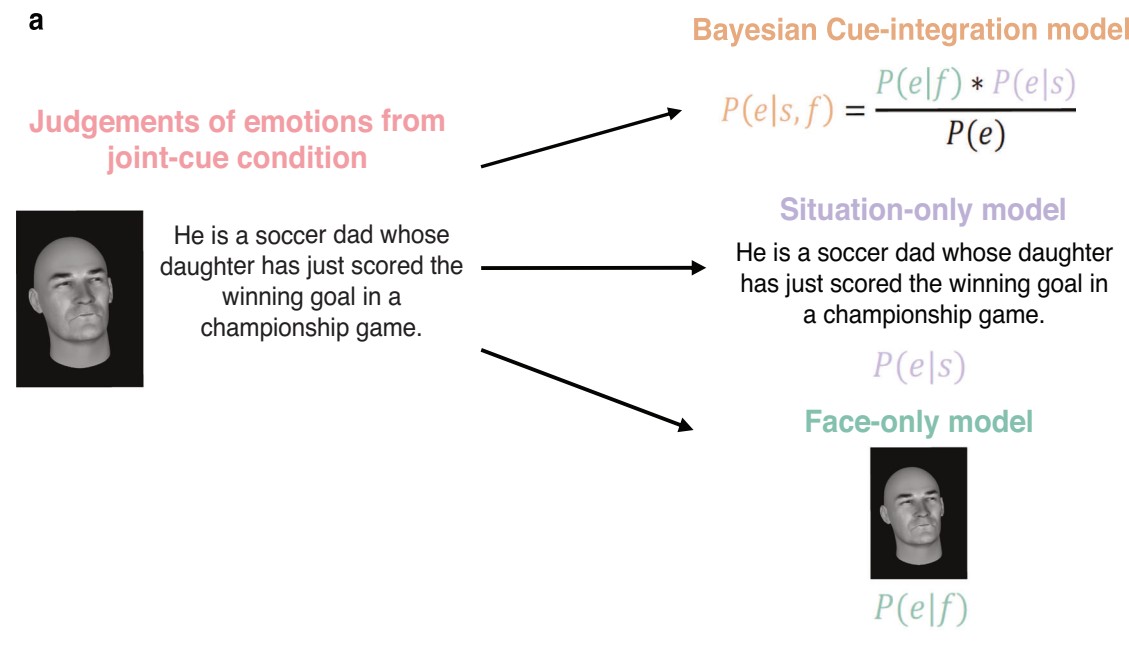

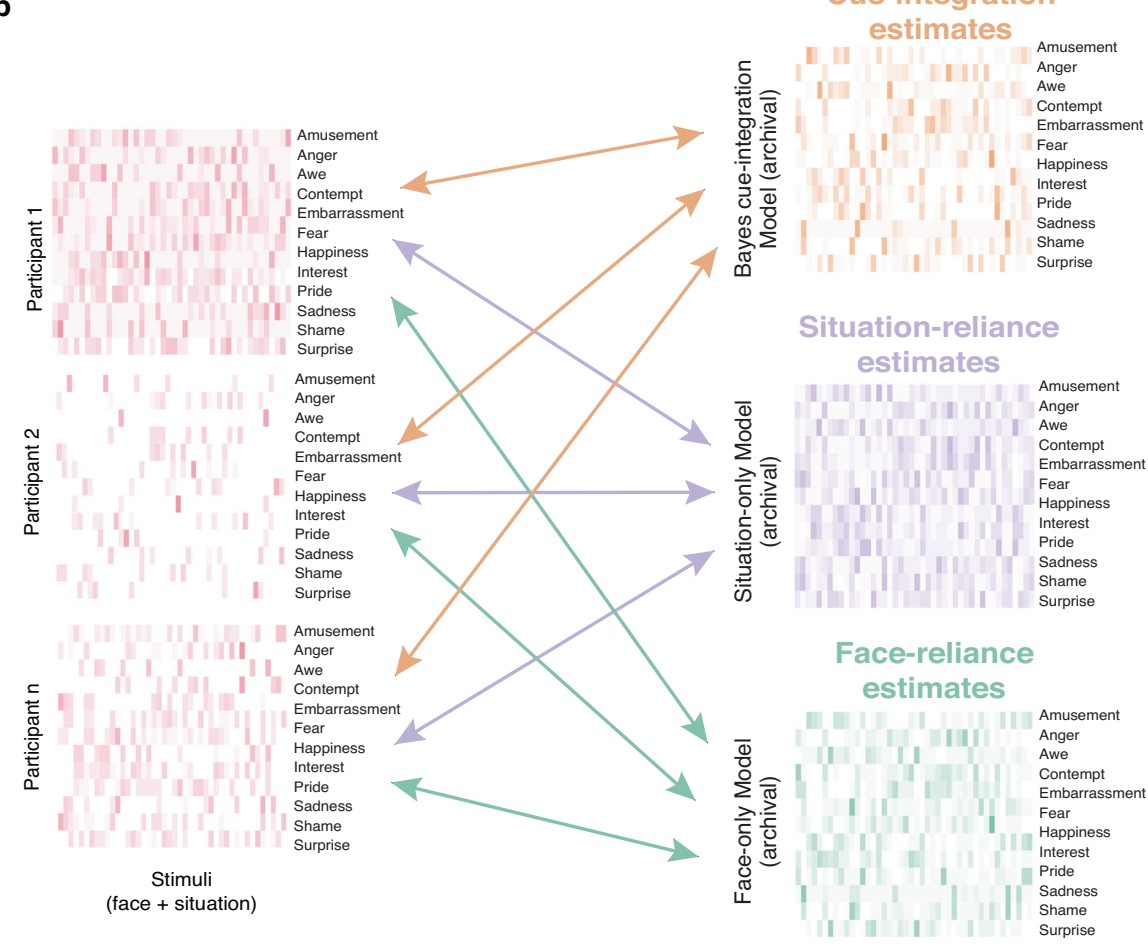

**Fig. 1 | Schematic representation of the analysis approach. a** Overall performance of each model is estimated by comparing average judgments of emotions for each stimulus in the joint-cue condition with Bayes cue-integration, Situation-only, and Face-only model estimates separately. **b** Each individual's reliance on cue-integration, situational and, facial cues was computed by comparing their judgment of emotions from the joint-cue condition with group level estimates of Bayes cue-integration, Situation-only and, Face-only model respectively. The group level estimates were obtained from the archival dataset for a subset of stimuli. *Note:* We are unable to provide example images of the sourced facial portrayals because the images are under copyright. The face image here is generated using FaceGen software (Singular Inversions, Inc., Toronto, ON, CA) based on human coded facial action units for the actors' portrayed facial expression.

facial and situational cues to the three normative model predicted estimates (Fig. 1b).

We compute Face-only, Situation-only, and Bayes cue-integration model estimates in Study 1 and compare it with people's judgments of emotions from viewing both facial and situational cues across Studies 1–5 (empirically collected for Studies 2–5). As a robustness check we collected an additional set of ratings for face-only and situation-only conditions to ensure that sampling variation did not significantly impact these model fits. We find that the distributions of ratings are highly comparable (see Supplementary Figs. S1–S3), and the model fits are consistent using the additional rating set (see Supplementary Tables S1).

### Do emotion inferences reflect Bayesian cue-integration?

In Study 1, we examined the robustness of the Bayesian model for cue-integration[31] using a more diverse and complex set of stimuli. Further, we examined the applicability of the cue-integration model to specific emotion categories (e.g., anger, fear, etc.). In this study, we used an archival dataset[24] and collected an experimental dataset (using the *priors task*). The archival dataset consisted of participants' ratings of descriptions of social scenarios (situation-only condition, $N = 839$), ratings of actors' portrayals of those scenarios (face-only condition, $N = 842$), and ratings of the combination of situations and faces (joint-cue condition, $N = 845$). Each participant provided ratings on 13 emotion categories (*amusement, anger, awe, contempt, disgust, embarrassment, fear, happiness, interest, pride, sadness, shame, surprise*) for a random subset of approximately 30 stimuli obtained from the larger set of 604, in one of the three conditions. We computed the Face-only $P(e|f)$ and Situation-only $P(e|s)$ model probabilities by averaging ratings across people for each emotion category and stimulus and then normalizing these across emotions for each stimulus such that the sum of probabilities across emotion categories for each stimulus equals 1.

To compute the Bayes cue-integration model probabilities $P(e|s,f)$, we require an estimate of people's prior expectations of inferring emotions $P(e)$ in addition to the above-mentioned Face-only and Situation-only model estimates. We collected this *priors data* by asking participants ($N = 45$) to rate their likelihood of perceiving each of the 13 emotion categories in their everyday lives. We computed the prior probability of emotions $P(e)$ by normalizing averaged ratings for each emotion category such that the sum of prior probabilities across the thirteen emotion categories equals 1. Then, we computed the Bayes cue-integration model probability estimates for each stimulus and emotion category by inserting the Face-only model probability, Situation-only model probability and emotion prior probability in Eq. 1. The cue-integration model probabilities were also normalized such that the sum of probabilities across emotion categories for each stimulus equals 1.

We first examined whether the cue-integration model best captures people's inference of emotions from these more diverse and complex facial and situational cues. We compared each of the three model estimates (Face-only, Situation-only, and Bayes cue-integration) to the empirical cue-integration probabilities that were computed by averaging the ratings for each stimulus in the joint-cue condition (empirical cue-integration). Two-tailed Pearson correlations of normalized empirical cue-integration probabilities with the model probabilities indicate that although each of the three models had a significant and strong correlation with empirical judgments, the Situation-only model had the highest correlation ($r$ (7850) = 0.865, $t = 153.04$, $p < 0.001$, bootstrapped 95% CI: 0.857, 0.873), followed by the Bayes cue-integration model ($r$ (7850) = 0.840, $t = 137.15$, $p < 0.001$, bootstrapped 95% CI: 0.830, 0.849), and finally the Face-only model ($r$ (7850) = 0.660, $t = 77.942$, $p < 0.001$, bootstrapped 95% CI: 0.644, 0.677) (Fig. 2a). While there was a statistically significant difference in the Situation-only and Bayes cue-integration model correlations ($t$ (7850) = −7.19, $p < 0.001$) based on a two-tailed test of difference

between two correlations, the difference in correlation values was relatively small suggesting similar model fit for integration and situation reliance. Compared to the Face-only model, we also observed a statistically significant difference between both the Situation-only ($t$ (7850) = 36.31, $p < 0.001$) and Bayes cue-integration ($t$ (7850) = 48.35, $p < 0.001$) models based on a two-tailed test of difference between two correlations.

It is possible that the Bayes cue-integration model performed significantly less robustly than the Situation-only model because people do not integrate priors over emotions when making inferences. To test this, we computed the Bayes-cue integration model using a flat prior, i.e., setting the priors to a uniform distribution. This also helps to examine whether the empirically collected priors are informative and improve model fit over a flat prior. The Bayes cue-integration model with flat priors correlates with empirical data ($r$ (7850) = 0.832, $t = 133.12$, $p < 0.001$, 95% CI: 0.826, 0.839) significantly less compared to the model with informative priors ($t$ (7850) = 9.13, $p < 0.001$) based on a two-tailed test of difference between two correlations. This suggests that global priors of emotions are informative and add value to the cue-integration model and that the Situation-only model does not perform better due to a lack of integration of priors.

The root-mean-squared-error (RMSE) for model predictions corroborated the correlation-based results. The Situation-only model had the lowest RMSE value (0.042, bootstrapped 95% CI: 0.041, 0.044) indicating the best-fit to the empirical data, followed by the Bayes cue-integration model (0.071, bootstrapped 95% CI: 0.068, 0.073), and finally the Face-only model (0.075, bootstrapped 95% CI: 0.073, 0.077) (Fig. 2b). These results suggest that the Situation-only model may better account for perceiver inferences.

We replicated these overall findings in Studies 2–4, such that we observe that, on average, the Situation-only model had a statistically higher correlation with people's judgments from both facial and situational cues (see Supplementary Table S2) compared to the other two models. This indicates that perceivers rely more heavily on situational information, even when given the opportunity to integrate, possibly as facial information adds little to no value beyond the knowledge of the context. As a robustness test, we also computed model estimates using direct certainty judgments instead of intensity judgments and find that strong reliance on situational cues is not a limitation of our approach of computing frequency-based probability estimates (see Supplementary Fig. S4). We find that the Bayesian model fit, and the Situation-only model fit, are not statistically significantly different when using certainty judgments ($t$ (570) = 1.06, $p = 0.289$) based on a two-tailed test of difference between two correlations. We therefore continue to present probabilities computed using the intensity judgments, but note that decisions reflecting probability matching behavior may not always apply to emotion inferences[38]. To accurately model emotion judgments, it is important to understand the underlying decision processes that may reflect different representations across different perceptual tasks[38].

### Does Bayesian cue-integration depend on the emotion inferred?

In addition to these overall patterns, we also compared how model predictions correlate with empirical judgments for each emotion category separately. The overall patterns were largely consistent across emotion categories, but there was also variability across emotions (Fig. 3). We compared confidence intervals computed for bootstrapped samples of difference in correlation values between the three models for all emotion categories[66]. The Situation-only model and the Bayes cue-integration model correlations were systematically and significantly higher than the Face-only model across all emotions except *Amusement* for which the Situation-only model did not statistically differ from the Face-only model. Further, the Situation-only model correlation was not uniformly statistically higher than the Bayes cue-integration model correlation across emotions. Compared to the

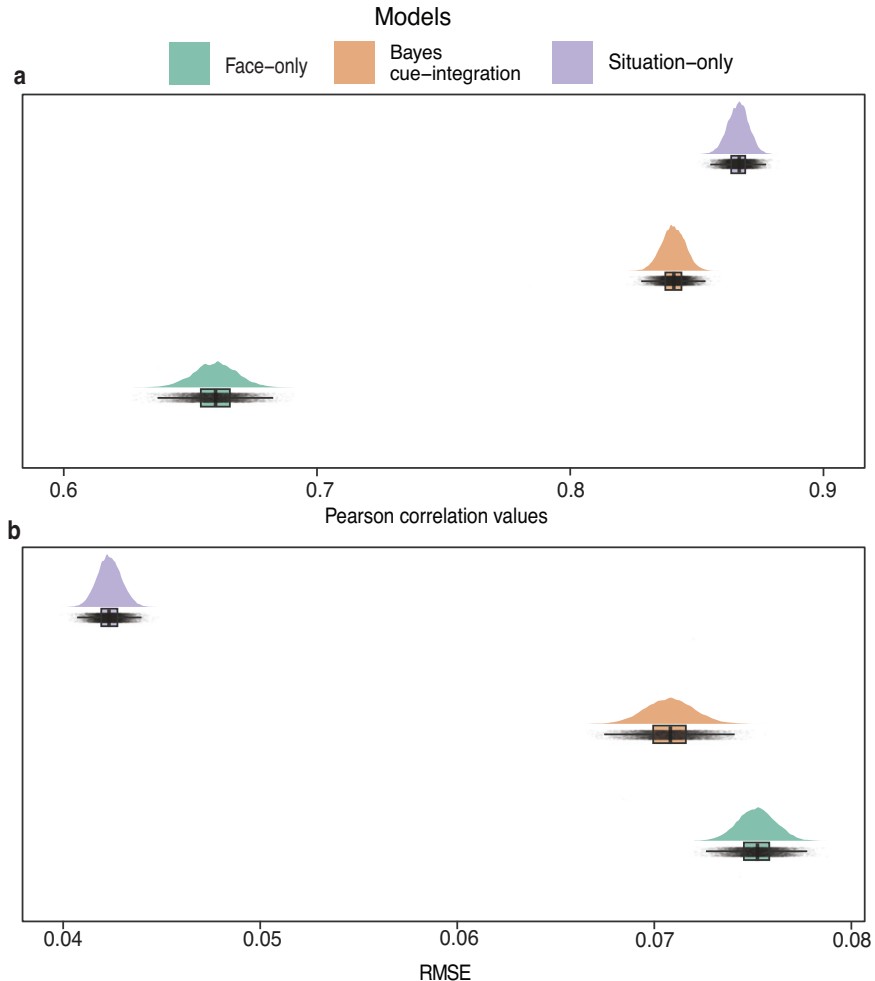

**Fig. 2 | Overall comparison of the three models in Study 1. a** Distribution of bootstrapped overall Pearson correlation values for each model derived from $n = 7852$ pairs of observations. Higher mean correlation values suggest better model fit. **b** Distribution of bootstrapped overall root-mean-squared-error (RMSE) estimate for each model derived from $n = 7852$ pairs of observations. Lower mean RMSE values suggest better model fit. The center line in the boxplot represents the mean correlation and mean RMSE values for the bootstrapped sample. The upper and lower limits of each boxplot represent the upper (75th percentile) and lower (25th percentile) quartiles of correlation or RMSE values for each model and the whiskers extending from the box represent the 1.5x interquartile range.

Bayes cue-integration model, the Situation-only model correlation was significantly lower for the emotion categories of *Amusement, Happiness* (Fig. 3a)*;* did not statistically differ for the emotion categories of *Anger, Contempt, Disgust, Sadness,* and *Surprise* (Fig. 3b); and was significantly higher for emotion categories of *Awe, Embarrassment, Fear, Interest, Pride, Shame* (Fig. 3c).

In sum, in Study 1 we find that when using more diverse and complex situation descriptions and facial portrayals the Bayes cue-integration model captures the empirical data well but does not consistently outperform the Situation-only model for overall emotion inferences. This suggests that when people have access to rich situational context, they tend to primarily rely on this information to infer emotions in others. However, this broad pattern does vary across emotion categories. When inferring the emotions of *Amusement* and *Happiness*, facial cues appear to be integrated with situational cues as suggested by the better fit of Bayes cue-integration model. Consistent with our predictions, this pattern of findings coincides with the variability in frequency of encountering emotion expressions. The emotion category where facial cues were integrated with situational cues, *Happiness*, is also reported as the category for which expressions are encountered most frequently in everyday life[40,67] (see Supplementary Tables S4 and S5). This suggests that base-rates of encountering expressions of emotions likely influences people's beliefs about the links between expressions and emotions, which in turn contributes to how facial cues are integrated into people's inferences.

### How do individuals vary in cue reliance and integration?

We extended the investigation of cue-reliance (reliance on facial or situational cues) and cue-integration to examine individual differences across three studies (Studies 2–4). To that end, in each of the three studies, participants rated pairs of facial portrayals and their corresponding situational descriptions (empirical joint-cue ratings) for the 13 emotion categories. The stimuli pairs were a randomly selected subset from the larger set ($n = 44$). This task was identical to the joint-cue condition from Le Mau and colleagues' paper[24]. We first computed each individual's empirical joint-cue probability estimates by averaging their emotion rating for each stimulus and normalizing the averaged ratings such that the sum of ratings across emotions for each participant and stimulus equals 1. These individual joint-cue probability estimates were then compared to the Face-only, Situation-only, and Bayes cue-integration model probabilities (obtained from Study 1 for the subset of 44 stimuli) to compute estimates of face-reliance, situation-reliance, and cue-integration respectively for each individual. We found considerable variation in the degree to which people relied on facial cues (Study 2: $M = 0.46$, $SD = 0.16$; Study 3: $M = 0.53$, $SD = 0.14$; Study 4: $M = 0.62$, $SD = 0.18$), situational cues (Study 2: $M = 0.60$,

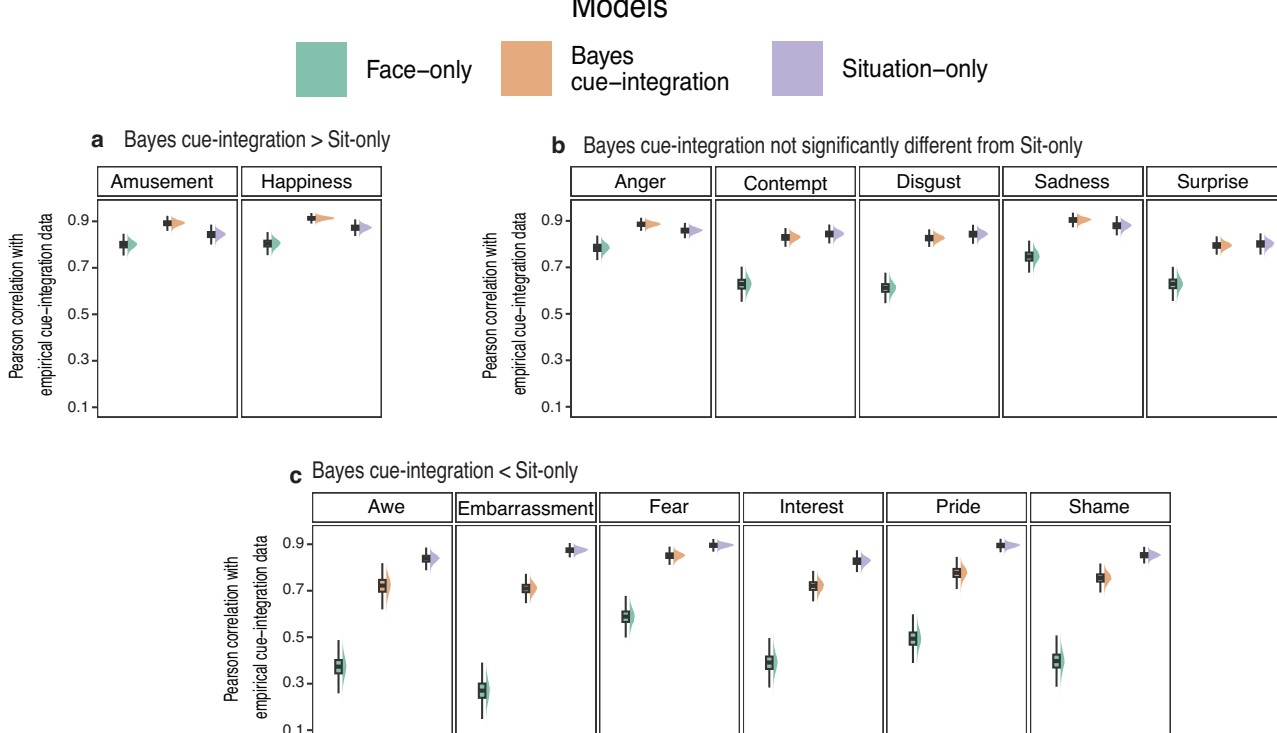

**Fig. 3 | Model correlation distribution for each emotion category.** Distribution of bootstrapped Pearson correlation of the models with the empirical judgments from joint-cue condition for emotion categories derived from $n = 604$ pairs of observations. Data is presented where the correlation of the Bayes cue-integration model with participant judgments was **a** significantly higher than that of the Situation-only model; **b** not significantly different from the Situation-only model and; **c** significantly lower than the Situation-only model. The center line in each boxplot represents the mean correlation for the bootstrapped samples. The upper and lower limits of each boxplot represent the upper (75th percentile) and lower (25th percentile) quartiles of correlation values for each model and the whiskers extending from the box represent the 1.5x interquartile range. See Supplementary Table S3 for values of 95% confidence intervals for difference in correlation between pairwise comparison of the three models for each emotion category.

$SD = 0.21$; Study 3: $M = 0.73$, $SD = 0.20$; Study 4: $M = 0.90$, $SD = 0.24$), and integration of facial and situational cues (Study 2: $M = 0.52$, $SD = 0.16$; Study 3: $M = 0.66$, $SD = 0.19$; Study 4: $M = 0.75$, $SD = 0.22$) across all three studies (Fig. 4a–c). Further, people also varied in whether they primarily relied on facial cues, situational cues, or integration of cues. These groups of individuals based on the highest performing model are visualized in Fig. 5a–c. In Study 2, though a majority of the participants (58.78%) relied primarily on situational cues alone, some (18.32%) relied primarily on facial cues alone, while others (22.9%) relied primarily on integration of both cues (Fig. 5a–c). The two subsequent studies (Studies 3 and 4) also demonstrate variation across people in whether they primarily relied on situational cues (Study 3: 63.16%, Study 4: 76.73%), facial cues (Study 3: 6.58%, Study 4: 3.63%), or cue-integration (Study 3: 30.26%, Study 4: 19.64%). For extended analysis on age-related effects on model estimates see supplementary materials (Table S6 and the accompanying note). Together, these findings suggest that we should be cautious in assuming that a single model, such as the one described in Study 1, sufficiently captures emotion inferences across individuals.

Next, we examined how individuals' face-reliance, situation-reliance, and cue-integration estimates relate to other measures of individual variation in the emotion domain (Studies 2–4). For each of these analyses, first we performed Fisher's r-to-z transformations of the cue-reliance and cue-integration correlation estimates. Then we computed two-tailed Pearson correlations between these estimates and self-report measures of individual variation in emotions adjusting for multiple corrections, such that all p-values reported are Bonferroni adjusted. We also removed multivariate outliers using the Mahalanobis distance method before computing correlations. We first examined whether people's cue-reliance and cue-integration estimates were related to their beliefs about their emotion differentiation ability (differentiation subscale of RDEES[52]). We did not find a statistically significant association between people's beliefs about their ability to differentiate between emotion experiences with their cue-reliance (Situation-reliance: $r (127) = -0.141$, $t = -1.609$, $p = 1.00$, 95% CI: $-0.307$, $0.032$; Face-reliance: $r (127) = 0.029$, $t = 0.327$, $p = 1.00$, 95% CI: $-0.145$, $0.201$) or cue-integration estimates ($r (127) = -0.034$, $t = -0.388$, $p = 1.00$, 95% CI: $-0.21$, $0.139$). The RDEES reflects individuals' global beliefs about how well they differentiate between various emotion experiences rather than their measured ability to differentiate[68]. As such, these beliefs about differentiating one's own emotion experiences may not track with individual differences in cue utilization and integration in emotion inference. Given the statistically insignificant relationships, we did not include this measure in subsequent studies (Studies 3 and 4).

Next, we examined whether participants' ability to detect presence and absence of cues in situational contexts (assessed using the CSI scale[46]) is related to cue-reliance. We hypothesized that people's utilization of situational cues to infer other's emotions would be positively associated with their ability to detect the presence and absence of contextual (or, situational) cues. We did not find credible evidence for a statistically significant relationship of situation-reliance with ability to detect absence of situational cues in two samples (Study 2: $r (127) = 0.067$, $t = 0.779$, $p = 1.00$, 95% CI: $-0.105$, $0.239$; Study 3: $r (146) = 0.023$, $t = 0.277$, $p = 1.00$; 95% CI: $-0.139$, $0.184$) but continued to include this measure across studies given the conceptual relevance to usage of situation information. In a final high-powered study ($N = 268$, Study 4), we found a small positive association between

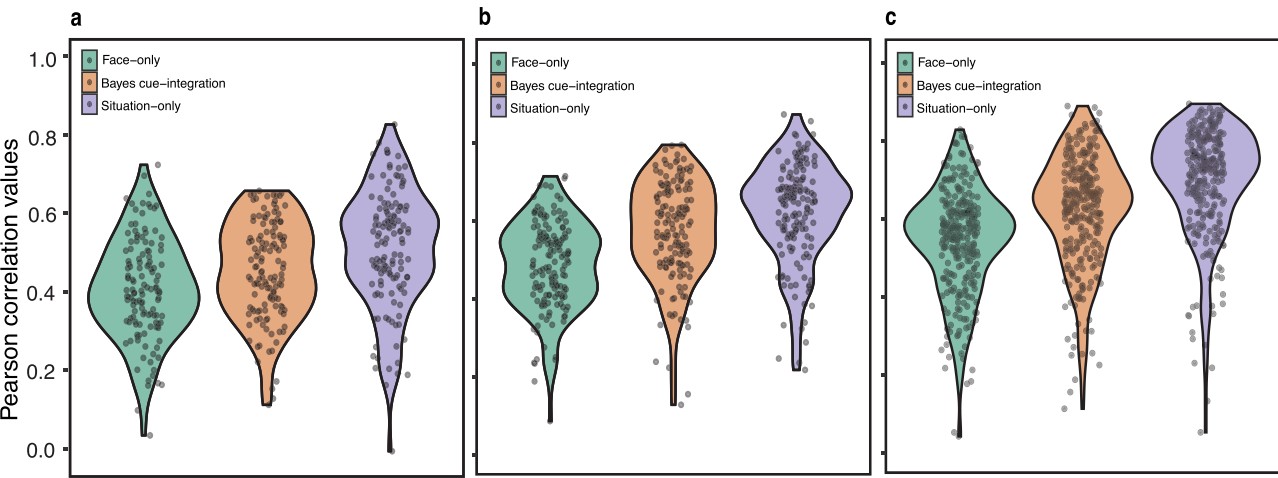

**Fig. 4 | Distribution of correlation values between model and participants.** Distribution of Face-only, Situation-only and Bayes cue-integration model correlation across individuals in **a** Study 2 ($N = 129$); **b** Study 3 ($N = 148$) and **c** Study 4 ($N = 268$).

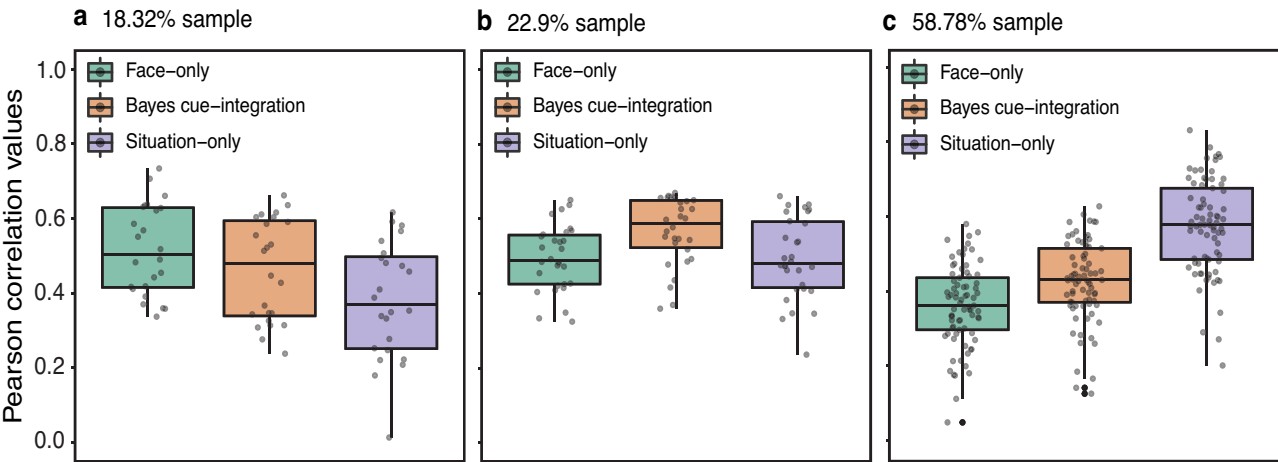

**Fig. 5 | Distribution of model correlation values grouped by participant's highest performing model.** Distribution of Pearson correlation values for Face-only, Bayes cue-integration, and Situation-only model for the group of participants who had the highest correlation for **a** Face-only model ($N = 24$; 18.32%); **b** Bayes cue-integration model ($N = 30$; 22.9%) and **c** Situation-only model ($N = 77$, 58.78%) respectively in Study 2. The error bars represent standard deviation. The center line in the box plots represents the median correlation value for each condition. The upper and lower limits of each boxplot represent the upper (75th percentile) and lower (25th percentile) quartiles of correlations values for each condition and the whiskers extending from the box represent the 1.5x interquartile range.

situation-reliance estimates and sensitivity to absence of contextual cues (Study 4: $r(266) = 0.189$, $t = 3.141$, $p = 0.030$, 95% CI: 0.071, 0.302), measured by the total score on the cue absence dimension of CSI. This suggests that people who were more likely to rely on situational cues when inferring emotions from both facial and situational cues were also more sensitive to detect absence of cues in situational context. However, this relationship was not consistently found in previous studies (Studies 2 and 3) that were powered to detect a comparable effect size, suggesting that we should cautiously interpret this effect. Further, we did not find a statistically significant relationship between situation-reliance estimates and sensitivity to presence of contextual cues as measured by the total score on the cue presence dimension of CSI across the three studies (Study 2: $r(127) = 0.048$, $t = 0.546$, $p = 1.00$, 95% CI: −0.126, 0.219; Study 3: $r(146) = 0.177$, $t = 2.177$, $p = 0.497$, 95% CI: 0.016, 0.329; Study 4: $r(266) = 0.108$, $t = 1.779$, $p = 1.00$, 95% CI: −0.012, 0.225). Given the relevance of CSI in understanding situational context, we did not hypothesize any relationship between CSI and face-reliance or cue-integration. We did not find credible evidence for a statistically significant association of face-reliance with detecting presence (Study 2: $r(127) = 0.084$, $t = 0.950$, $p = 1.00$, 95% CI: −0.090, 0.253) or absence (Study 2: $r(127) = 0.041$, $t = 0.458$, $p = 1.00$, 95% CI: −0.133, 0.212) of contextual cues, nor people's cue-integration estimates with detecting presence (Study 2: $r(127) = 0.107$, $t = 1.216$, $p = 1.00$, 95% CI: −0.067, 0.275) or absence (Study 2: $r(127) = 0.035$, $t = 0.400$, $p = 1.00$, 95% CI: −0.138, 0.207) of contextual cues. The weak but inconsistent association we did document is perhaps not surprising given that the CSI is a relatively narrow measure of context sensitivity. It measures people's ability to detect presence and absence of contextual cues related to threat (e.g., urgency to respond, control by self and others). The scenarios in our studies likely incorporate a much broader range of contextual features beyond those relevant to threat (e.g., interaction with others, and consistency with goals), as reflected in the broad range of emotions they are associated with (see Supplementary Data 1).

To address the potentially limited focus of CSI, in Studies 3 and 4 we examined whether situation-reliance tracked with a broader ability-based measure that captures individual differences in understanding the links between emotions and situations. This was

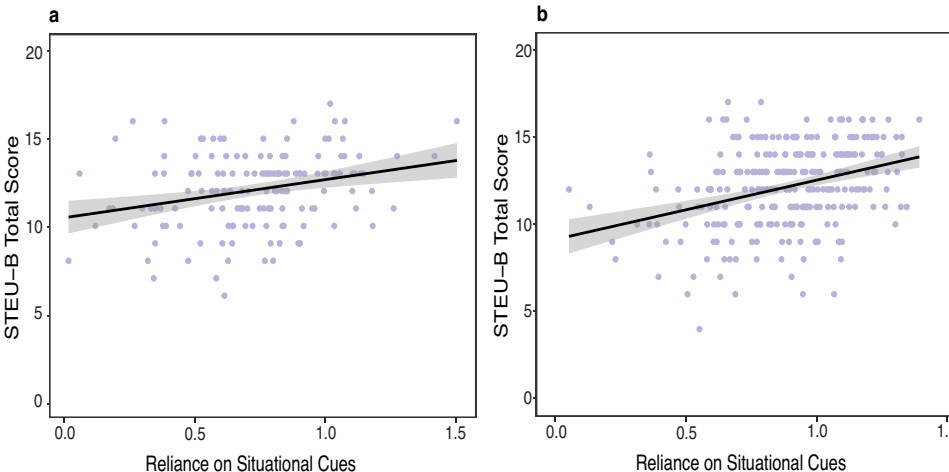

**Fig. 6 | Relationship between reliance on situation cues and situated understanding of emotions.** Moderate positive relationship between individual's Situation-only model estimates (x-axis) and their situated understanding of emotion experiences (y-axis) in **a** Study 3 (N = 148) and **b** Study 4 (N = 268). Each point on the plot represents an observation from the dataset. The black line represents the line of best-fit for a linear relationship between the variables and the gray shaded band represents a pointwise 95% confidence interval on the fitted values.

assessed using the brief version of the Situated Test of Emotion Understanding (or STEU-B[41]). We hypothesized that the variability in situation-reliance estimates would be positively associated with people's situated understanding of emotions—that is, their ability to accurately identify (based on group consensus) which emotions are most likely experienced by people in different situations. Compatible with our predictions, we found a reliable and statistically significant positive association between people's understanding of situated emotions and their situation-reliance estimates (Study 3: $r$ (146) = 0.281, $t$ = 3.533, $p$ = 0.009, 95% CI: 0.125, 0.423; pre-registered replication Study 4: $r$ (266) = 0.343, $t$ = 5.949, $p < 0.001$, 95% CI: 0.232, 0.444) suggesting that people who rely more on situational cues when inferring emotions are also able to better understand the links between emotions and situations (Fig. 6a, b).

In addition, we examined the temporal stability of individuals' face-reliance, situation-reliance, and cue-integration estimates (Study 5). Following the analytical technique used in Studies 2–4, we first computed individual's face-reliance, situation-reliance, and cue-integration estimates for both sessions one and two and transformed those estimates using Fisher r-to-z transformation. We then computed two-tailed intra-class correlations to examine the consistency of cue-reliance (both face and situation) and cue-integration estimates for individuals across time. Results indicated that on average there is a moderate test-retest reliability for reliance on situational cues alone ($ICC$ = 0.593, $F$ (109, 109) = 3.136, $p < 0.001$, 95% CI: 0.216, 0.768) and integration of cues ($ICC$ = 0.672, $F$ (109, 109) = 3.260, $p < 0.001$, 95% CI: 0.508, 0.779) but weak reliability for reliance on facial cues alone ($ICC$ = 0.370, $F$ (109, 109) = 2.436, $p < 0.001$, 95% CI: −0.170, 0.656) (Fig. 7). This suggests that people's reliance on situational cues and their integration of facial and situational cues is moderately stable over time. However, people's reliance on facial cues is not a stable individual difference across time. We also checked the robustness of these results after removing multivariate outliers and found similar results for temporal stability of individual differences in cue-reliance and cue-integration (see Supplementary Table S7).

## Discussion

In our work, we examine people's utilization of facial and situational cues to infer emotions in others. Inferences based on a single cue (face or situation) and integration of cues were compared to people's judgments of emotions given access to both facial portrayals and situational context. In addition to the group level estimates, we examined variability in utilization of facial and situational cues across people and emotion categories.

Our data shows that overall inferences of other's emotion states based on access to their facial and situational cues were well aligned with both inferences based on situational cues alone and integration of cues. On average, people's emotion inferences based on situational cues alone had a significantly better fit, but only slightly, with joint-cue empirical judgments compared to inferences based on Bayesian cue-integration. Further, compared to the situation-only and integration models, we found inferences based on facial cues alone were consistently and significantly weaker in capturing emotion inferences made in presence of both faces and situations. One interpretation is that, in our study context, the information that facial expressions contain are a (small) subset of the information in the situation context, such that face-only inference is poor, and cue-integrated inferences do not differ much from situation-only inferences. This contradicts one aspect of the *common view:* that facial behaviors are tightly linked to an emotion state, such that access to facial cues should provide diagnostic information about the underlying emotion. These findings suggest instead that people's inferences about what someone else is feeling are adequately explained by their beliefs that situational context leads to emotion experience. These results additionally point to the need for a richer lay theory of emotions: perceivers may in fact consider expressions to be situated (i.e., in the causal model, adding a causal link between situations and expressions)[69].

The observed strong reliance by perceivers on situational context when inferring emotion goes beyond past findings demonstrating strong situational reliance (for review, see refs. 15,25). We demonstrate that situations routinely take precedence in emotion inference rather than being integrated with facial portrayals. This finding emerged in our work which avoided the pitfalls of contrasting the meaning of situational and facial cues. Critically, our findings add to existing research examining perceivers' inferences from spontaneous expressions derived from real-world contexts. For example, previous research demonstrates that spontaneous real-world facial movements in high-intensity contexts of sporting wins and losses (when separated from body language cues) have no utility in discriminating valence-based information[70]. Our findings further indicate that non-canonical facial portrayals capturing a broader range of intensities and emotions are not only similarly less informative for emotion inferences but that they are rarely integrated with the situational context (conceptually replicating[24]).

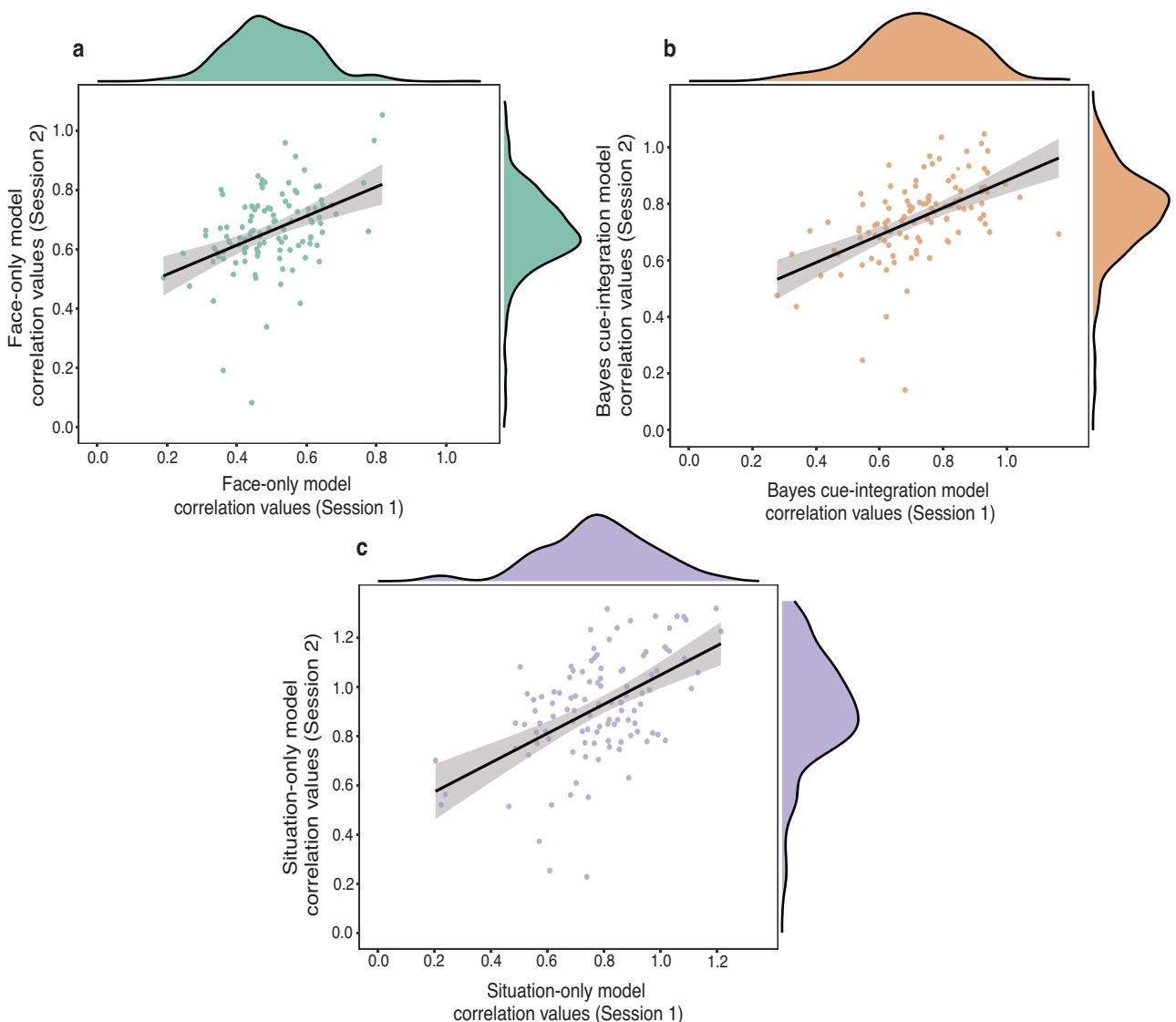

**Fig. 7 | Test-retest reliability for cue-reliance and cue-integration estimates.** Relationship between Fisher r-to-z transformed model correlation values for the two-time points for **a** Face-only model; **b** Bayes cue-integration model and **c**, Situation-only model. The Pearson correlation values were Fisher r-to-z transformed before computing the intra-class correlations. The black line represents the line of best-fit for a linear relationship between the variables and the gray shaded band represents a pointwise 95% confidence interval on the fitted values. The same sample ($N = 110$) was tested at session 1 and session 2.

The reliance on situation context may be particularly relevant when thinking about real-world emotion inferences. The cues used in our work have greater diversity and complexity. For example, portrayals of emotion in this dataset rarely conformed to a priori canonical expectations for how faces move[24]. For instance, it was rare that in situations that would be viewed as interest-inducing, actors portrayed that emotion by narrowing their eyelids and producing a controlled smile (the proposed canonical expression). This could be because the actors were asked to enact the complexity of situations from the descriptions instead of providing them with an emotion category label or predetermined facial muscle movements to enact (as is common in the literature). Indeed, there was variability in how actors used their faces to portray the emotions. However, these stimuli do not sufficiently capture the spontaneous naturalistic expressions that arise in everyday social situations. These findings therefore underscore the importance of testing theories and computational models of emotions using diverse sets of more naturalistic stimuli. Such stimuli can help test people's ability to infer emotions in a capacity that is more closely related to the real-world phenomenon.

Our findings also parallel prior work examining the role of facial information in impression formation and updating. Research on trait inferences suggests that people infer traits from appearance-based cues from faces (for example, see ref. 71) especially when asked to make speeded judgments (for example, see refs. 72,73). However, both explicit[74] and implicit[75] impressions of traits based on facial information are updated in the presence of explicit descriptions of target behaviors/characteristics. Together with our results, findings such as these[74,75] suggest that the assumption of facial dominance in social cognition may be overstated. Indeed, prior research suggests that perceivers overestimate the value of facial information over context in forming inferences about emotion[76]. People's inferences of others, be they trait or state inferences, can be formulated based on facial information as countless studies have shown, but appear to be heavily constrained by other sources of information such as knowledge of the context.

The overall pattern of findings was relevant for most emotion categories measured in our work but, there was some variation across emotion categories in whether people relied relatively more on the situational context. When inferring emotions of *Amusement* and *Happiness*, people on average were most likely to integrate facial and situational cues as demonstrated by the better fit of our data to the Bayes cue-integration model. This suggests that when inferring these emotions, facial cues were likely perceived to be informative, alongside the situational cues and were therefore more likely to be integrated.

The variation observed in cue-integration for inferring different emotion categories coincides in meaningful ways with variation in encountering and processing facial expressions in people's everyday lives (for example, see refs. 40,67). The emotion category of *Happiness*, where people were most likely to integrate facial cues with situational cues, is also the category for which people have a higher perceptual sensitivity for[77,78] and report seeing the most facial expressions of in their everyday life[67]. Similarly, the emotion category of *Fear* that people report seeing the least facial expressions of (among those studied) is among the categories where people were not significantly more likely to rely on integration. These findings compliment previous work showing that greater frequency of encountering facial expressions for an emotion category is also linked to greater recognition "accuracy" from canonical expressions and reduced latency to respond[67]. Together, our findings suggests that though information from facial portrayals is sometimes integrated into inferences of emotions, this integration varies in meaningful ways across categories. This variation has been rarely attended to when examining how people infer other's emotion states but potentially holds implications for determining relevance of cues in situated instances of different emotion categories.

We also demonstrate variability across individuals in their tendency to rely on situational cues, facial cues, or to integrate these cues when inferring emotions in others. Across three studies, we find that a majority of individuals infer other's emotions primarily based on situational cues, but some individuals infer based on integration of facial and situational cues while a small percentage of individuals' inferences are based on primarily the facial cues. This variation suggests that the strength of lay beliefs about emotions being caused by situations or leading to expressive behaviors can vary across people. Such variation across individuals is often ignored in cognitive modeling although there is some limited work examining 'individual differences' by grouping individuals on specific model parameters[79,80]. Here, we focus on modeling individual differences in cue utilization and integration with a consensus-based approach. An alternative would be to use reports of the emotion experienced or expressed by the target (for discussion see ref. 81). It is possible that these two approaches may yield distinct results insofar that the shared cultural model captured by consensus judgments diverges from the expression-emotion links for a given target. Further, in our work, we find that people's reliance on situational cues to infer emotions was reliably linked to their ability to understand the situated nature of emotions. That is, people who were better at understanding the links between emotions and situations also relied more on situational cues to infer emotions. This further suggests that inferring emotions based on situational context holds validity and varies across individuals in systematic ways based on their ability to make use of contextual information.

This finding is broadly consistent with prior work demonstrating that people's ability to understand the links between emotions and situations is positively associated with their ability to identify other's intentions, beliefs, and feelings in social situations[82] and their ability to switch between different emotion regulation strategies depending on the demands of different situations (commonly called regulatory flexibility and adaptability)[83]. These associations imply that the extent to which people rely on situational information is likely reflective of their normative ability to understand the situated nature of their own as well as other's mental states.

In addition to the variability across individuals, we also examined variability within individuals across time. Our findings suggest that over time people's reliance on situational cues and integration of cues are moderately stable tendencies. Such temporal stability suggests trait like tendencies indicating that variation in how individuals rely on situational cues and integration of cues are somewhat stable over time. In contrast, reliance on facial cues had low temporal stability. When examining the pattern of model fit estimates, we observe that facial cue reliance was generally higher at the second timepoint. This indicates that how people use facial cues to infer emotions may be contextual, such that familiarity drives up reliance on facial cues.

Our findings provide initial support for extending the examination of theories and computational models of emotions using diverse and complex social scenarios and facial portrayals, but more empirical work is required to address limitations of our approach. While the stimuli we employed here include more complexity and diversity than past work, they are not representative of the myriad of everyday experiences of emotion that occur. Further, the facial portrayals in the stimuli used are ultimately acted and do not capture actual spontaneous naturalistic expressions occurring in real-world contexts. Another limitation is that the situational cues here were all descriptions of social situations rather than dynamically unfolding situations as experienced in real-life. As such, the features of the situation that were highlighted for participants may not be as readily accessible in everyday life. Real-world understanding of situations requires the ability to represent abstract features and deploy attentional resources to prioritize relevant features (for example, see refs. 84,85). Mitigating this concern is the relatively low consensus for situational ratings in our data (see Supplementary Figs. S5–S7), which suggest that these cues have variable interpretations and are thus ambiguous. Further, even when individuals have incomplete or uncertain knowledge about a particular situation in everyday life, it is likely that aspects of the perceived physical environment still form the basis for social predictions. As a result, it is unlikely that many real-world instances of emotion perception are ever truly decontextualized. Testing cue-reliance and integration in different contexts with varying degrees of informational uncertainty is a question open for exploration. In addition, we use model estimates computed from Study 1 sample in Studies 2–4. This assumes that the inferences about emotion from these cues would be comparable across samples, which could be a limitation (see robustness checks in Supplementary Figs. S1–S3 and Supplementary Table S1). Finally, our samples used to examine individual-level variability in cue-usage and cue-integration were largely based on convenience samples obtained online. This approach limits the generalizability of our findings. We begin to address this concern with our final nationally representative sample (based on age, race, and gender), but there are many additional dimensions along which generalizability should be addressed.

In future work, it will be valuable to examine cue-integration using alternative theoretical assumptions that can be tested using Bayesian inference. Here we examine cue-integration stemming from the *lay intuition* that people expect emotion experiences to be arising from situational outcomes and leading to facial portrayals[31]. This *lay intuition* suggests that expressive behavior directly flows from emotions (which are themselves situated) such that the expressive behavior may not be further constrained by knowledge of the situational context. But people may hold a lay theory that emotional expression is situated (i.e., an additional causal link between situations and expressions in the DAG), suggesting that the current cue-integration model itself may have limitations. As an example, the non-verbal expression one might expect in an instance of anger at one's boss would be highly distinct from an expression of anger at a driver who cut you off in traffic. It would be fruitful to examine integration of facial and situational cues

based on an alternative model that assumes facial cues are constrained by both emotion and the situation. Testing this model will necessitate building a naturalistic stimulus set that contains sufficient and systematic situational variation within each emotion category to model this complexity. Another extension is to test the generalizability of the present findings across different types of cues, including when bodily cues serve as context. For example, based on prior work[70], it is likely that a bodily-context model would at least similarly outperform the Face-only model. The current modeling approach can be extended to formalize lay beliefs about a range of cues in people's understanding of emotions. The generalizability of our findings can also be tested by examining the extent to which reliance on single cues applies to varying degrees of contextual and facial information. An open question is whether people rely more on visual cues (e.g., faces or body language) in the presence of incomplete contextual information. For instance, when playing a game of poker with strangers at a casino, you do not have access to your opponent's cards. Such lack of context information may lead you to rely on visual cues to understand what the opponent might be feeling and use that to then reason about the context (e.g., do they have good or bad cards). Another future direction will be to examine whether there are emotion-specific individual differences, a question which was beyond the scope of the present study. In addition, in our work we examined some factors that may impact individual's utilization of situational cues, but future work can address similar questions about utilization of facial cues. People believe that different emotion categories are expressed variably in the face and body and these beliefs about material expressions are linked to beliefs about emotions having an innate essence[86]. Strong beliefs about material expression and innateness of emotions may lead to greater utilization of facial cues, and this would be valuable to examine in future work. Finally, an interesting future direction to pursue is a modeling approach that accounts for the profile of emotion ratings for a given stimulus, instead of treating each emotion for each stimulus separately when evaluating model fit. For instance, an experience of 'disgusted-anger', which may reflect moral-outrage[87], would not be treated the same for disgust and anger as separate emotions.

Overall, the present work reveals that people heavily weigh situational contexts when inferring what someone else is feeling. This challenges the *common view* that strongly emphasizes the role of facial cues in emotion inferences. Further, the extent to which situational context is weighed in emotion inferences varies across emotion categories and individuals in systematic ways that potentially have real-world social implications. These findings support arguments about emotion inferences being complex, variable and construed from multiple factors including knowledge of the context.

## Methods

All studies comply with the ethical guidelines for conducting human subjects research and were administered under the exempt protocol approved by the Yale University Institutional Review Board (IRB #: 2000028669). Participants across all studies provided informed consent before participating in the study and were compensated a pro-rated amount for the duration of task using the then Prolific suggested hourly rate marked as 'Good' ($10–12/h).

### Study 1
**Overview.** This study comprised of two datasets—(1) an archival dataset obtained from Le Mau and colleagues (2021), and (2) an experimental dataset obtained to compute prior expectations of inferring emotions.

**Archival Data.** The archival data[24] contain ratings of social scenarios, ratings of actors' portrayals of those scenarios, and ratings of the combination. These ratings were obtained for a total of 604 stimuli pairs (scenarios and poses) that were sourced from two books: *In*

*Character: Actors Acting*[50] and *Caught in the Act: Actors Acting*[51]. These volumes contain images of expressions posed by a pool of professional actors after they were provided with emotionally evocative scenarios. Some examples of the scenarios include: 'She is confronting her lover, who has rejected her, and his wife as they come out of a restaurant'; 'He is a motorcycle dude coming out of a biker bar just as a guy in a Porsche backs into his gleaming Harley'. We are unable to provide example images for the facial portrayals due to copyright restrictions.

The dataset included 75,390 observations by participants who rated a random subset of approximately 30 stimuli (out of a total of 604 stimuli). Each observation included ratings on 13 different emotion categories. Participants provided these ratings for one in 3 different conditions—*face-only* ($N = 842$), *situation-only* ($N = 839$), face and situation *combined* ($N = 845$). In the *face-only* condition participants viewed only the actor's portrayals of scenarios. In the *situation-only* condition, participants viewed only the description of those scenarios. In the *combined* or *joint-cue* condition, participants viewed both the description of scenarios along with the actor's portrayals of those scenarios. The 13 emotions they rated were—*amusement, anger, awe, contempt, disgust, embarrassment, fear, happiness, interest, pride, sadness, shame, surprise*. Each emotion category was first rated for presence, i.e., if the participant observed an emotion they respond saying *yes* or *no*. If the emotion was present, i.e., they responded *yes*, then the participant rated the intensity of that emotion on a 4-point-Likert scale (*slightly, moderately, strongly, intensely*)[88].

Actors' facial portrayals in this dataset did not reliably align with proposed canonical facial configurations of emotion categories[24]. Instead, the facial poses conformed with the variability observed in spontaneous emotion expressions in everyday life. For example, similar to people in their daily lives, actors scowled about 30% of the times when portraying scenarios consistent with the emotion anger. The complexity of this stimulus set, including the scenarios and the portrayals of them, offers greater ecological validity and range.

**Collecting data on people's prior expectations.** We collected rating data for people's perceptions of each of the above-mentioned 13 emotion categories in their everyday lives (*priors task*). It is standard in certain cognitive modeling paradigms to empirically collect priors by asking participants to rate a single question (for example, see ref. 89). In addition, we collected empirical priors rather than estimating priors based on the archival data directly because we do not assume the range of stimuli is representative of emotional instances in everyday life that likely inform laypeople's judgments. We compared these priors to similar ratings collected by Somerville and Whalen[40] (see Supplementary Table S8). Further, we included a short *rating task* that was identical to the joint-cue condition from Le Mau and colleagues' paper[24] to ensure consistency between our collected data and the archival data i.e., to examine whether providing such ratings would change the nature of judgments. If reporting on likelihoods changes the nature of emotion inference, this may undermine the proposal that these likelihood estimations inform emotion inference. We confirmed that the distribution of ratings for the subset of stimuli were similar in the *rating task* and the archival data for all 13 emotion categories.

**Participants.** 45 native English-speaking participants were recruited from the US (20 male, 25 female, mean age = 38-year, age range = 18–60 years) using the online data collection platform Prolific. We recruited 45 participants in the current sample to approximately match the archival dataset where on average ~40 participants responded to a given stimulus. Detailed demographic information is provided in Supplementary Table S9.

**Priors task.** In the priors task, participants responded to a question that asked about their likelihood of perceiving each emotion category in their daily lives on a 7-point-Likert scale with three anchor points—1:

not at all likely, 4: moderately likely, 7: extremely likely. The question stated: 'When you see people experiencing emotions in your day-to-day life, how likely are you to perceive people experiencing [Emotion]'.

**Rating task.** The task design was identical to the joint-cue condition in the archival data[24] such that each participant responded (*yes/no*; followed by *slightly, moderately, strongly, intensely*) to the 13 different emotions for a set of facial portrayals and situation descriptions. The stimuli set consisted of a random sample of 10 stimuli that was drawn from the larger pool of 604 stimuli. The order of stimuli was randomized.

**Procedure.** All participants read an online consent form before agreeing to participate in the Study. Participants then read instructions for the *rating task* where they would provide ratings for emotions after viewing descriptions of scenarios and actor's portrayals of emotions in those scenarios. The instructions were identical to those provided by Le Mau and colleagues[24]. This was followed by a question aimed to validate that they read the instructions. After answering this question, participants read instructions for providing rating on the likelihood of perceiving each emotion category and responded to another question to validate that they read the instructions. After answering these questions, they were given a quick reminder of the instructions. This was included so that participants, especially those who gave an incorrect response, could be reminded of the task instructions. Participants then provided ratings on the *priors task* followed by the brief 10-item *rating task*. At the end, participants also filled out a brief demographic questionnaire and were thanked and compensated for their participation.

**Data preparation.** We removed participants with spurious data from the archival (*N* = 13) and *priors task* data (*N* = 1) leaving a total of 2513 and 44 participants in the respective datasets for analysis.

## Study 2

**Overview.** Study 2 comprised of an experiment and two questionnaires (details below). This study had a within-subjects design with all participants responding to the experiment followed by the two questionnaires.

**Participants.** 150 native English-speaking participants between the age of 18–60 years were recruited from the US using the online data collection platform Prolific. To determine sample size, we carried out power analysis for bivariate correlations controlling for multiple comparisons using G*Power[90]. The results indicated that with a sample of 149 we have enough power (*1-β* = 0.8) to detect a small-medium effect size (*r* = 0.3). Eight people did not complete the study, leaving a total sample of 142 participants (Mean age = 33.84, SD age = 11.69; 50.7% female, 47.18% male and 2.11% non-binary; 76.05% White, 10.56% Asian, 7.75% Black or African American). Detailed demographic information is provided in Supplementary Table S10.

**Experimental task.** In the task, each participant viewed 44 stimuli pairs that consisted of images of facial portrayals alongside descriptions of social situations. For each stimulus, participants rated the presence (*yes/no*) and intensity (*slightly, moderately, strongly, intensely*) of 13 emotion categories, identical to the joint-cue condition in the archival data[24]. The 44 stimuli pairs were present in a random order and were randomly sampled from the larger set of 604 stimuli pairs used in the archival data[24]. The number of stimuli was determined using a priori power analysis with 'pwr' package in R[91] to obtain a medium effect size (*r* = 0.4). We chose a medium effect size for the stimuli as this was the smallest effect size for bootstrapped model correlations obtained from Study 1.

**Context-sensitivity index or CSI[46].** CSI is an ability-based measure that captures individuals' sensitivity to the presence and absence of contextual cues. The questionnaire consists of 6 short descriptions of situations that arise in everyday life. For each situation description, participants rate a set of 3–4 questions related to psychological features of that situation (e.g., control by self, urgency) on a 7-point scale with 1 indicating 'not at all' and 7 indicating 'very much'.

**Range and differentiation of emotional experience scale or RDEES[52].** RDEES is a self-report measure that captures individual differences in emotional complexity by measuring people's reported range of emotional experiences (*range* sub-dimension) and their reported propensity to make subtle distinctions within emotion categories (*differentiation* sub-dimension)[52]. The questionnaire consists of 14 items, divided into the two sub-scales, that are each rated on a 5-point scale with 1 indicating that the statement 'does not describe me very well' and 5 indicating that the statement 'describes me very well'.

**Procedure.** Participants first read and signed an online consent form agreeing to participate in the Study. Then, participants read instructions for the experimental task, these were identical to those provided by Le Mau and colleagues (2021). This was followed by a question aimed to validate that participants read the instructions. Participants then performed the experimental task followed by CSI and RDEES questionnaires. The order of the questionnaires was counterbalanced across participants. At the end, participants also filled out a brief demographic questionnaire and were thanked and compensated for their participation.

**Data preparation.** To correctly use the rating scale in the emotion inference task, participants were instructed to only rate intensity when the emotion was present. Accidental ratings on the intensity scale could not be undone, based on limitations of the platform. We instructed participants to rate 'slight intensity' when an accidental intensity rating was made. Fully incorrect ratings were defined as marking the absence for an emotion but rating its perceived intensity as anything higher than slight intensity. Participants who provided accidental, incorrect, or had missing ratings for more than 10% of their data were removed from analysis. Data from 11 participants was removed leaving a sample size of 131 participants. Incorrect ratings were removed on a trial-by-trial basis and accidental ratings were corrected to "emotion absent". We also performed multivariate outlier detection using the Mahalanobis distance method from the *Routliers* package in R[92]. 2 subjects were identified as outliers and their data was removed resulting in 129 participants for final analysis.

## Study 3

**Overview.** Study 3 was an extension of Study 2 and comprised of an experiment and two questionnaires (details below). This study had a within-subjects design with all participants responding to the experiment followed by the two questionnaires.

**Participants.** 168 native English-speaking participants between the age of 18–60 years were recruited from the US using the online data collection platform Prolific. To determine sample size, we carried out power analysis for bivariate correlations controlling for multiple comparisons using G*Power[90]. The results indicated that with a sample of 139 we have enough power (*1-β* = 0.8) to detect a small-medium effect size (*r* = 0.25). We oversampled for an additional 20% of data to account for exclusions and attrition based on previous study, resulting in a total sample of 168 individuals. Six people failed attention checks built in the task, so their data was removed from all analyses, leaving a sample of 162 individuals (Mean age = 36.04, SD age = 12.87; 59.88% female, 34.57% male and 5.56% non-binary; 71.6% White, 12.35% Black or

African American, 5.56% Asian). Detailed demographic information is provided in Supplementary Table S10.

**Experimental task.** The experimental task was identical to that used in Study 2.

**Context-sensitivity index or CSI**[46]. CSI was identical to that used in Study 2.

**Situational test of emotional understanding-brief or STEU-B**[41]. STEU-B is a short version of the STEU[45] that measures an individual's understanding of the link between emotions and situations. It consists of 21 items, each describing an emotional situation, accompanied by multiple-choice scale with 5 emotion words. Participants are asked to choose the emotion that is most likely a result of the situation described. The responses are scored as correct or incorrect based on the scoring guidelines determined by MacCann and Roberts (2008).

**Procedure.** The procedure of this study is identical to Study 2 except that RDEES was replaced by STEU-B.

**Data preparation.** Following the same procedure from Study 2, participants who provided accidental, incorrect, or had missing ratings for more than 10% of their data were removed from analysis. Data from 10 participants was removed leaving a sample size of 152 participants. We also performed multivariate outlier detection using the Mahalanobis distance function from the *Routliers* package in R[92]. 4 subjects were identified as outliers and their data was removed resulting in 148 participants for final analysis.

## Study 4
**Overview.** Study 4 was a pre-registered replication (https://osf.io/u9abg, date of registration: May 27, 2022) of our previous work with a nationally representative sample and comprised of an experiment and two questionnaires (details below). This study had a within-subjects design with all participants responding to the experiment followed by the two questionnaires. There were no deviations from the pre-registration, but some additional analysis was carried out in response to reviewer comments detailed in Supplementary Table S6.

**Participants.** A nationally representative sample of 303 participants from the US was recruited using the online data collection platform Prolific. The sample size was planned to provide a more stable estimate of previously observed effect sizes, as correlations tend to stabilize at $N = 260$[93,94]. We are recruiting more than 260 participants to meet the minimal sample size ($N = 300$) on Prolific to recruit a nationally representative sample. We also conducted an a priori power analysis for bivariate correlations to detect the effect size for the relationship between STEU-B scores and situation-reliance estimates that was observed in the previous study ($r = 0.3$), controlling for multiple comparisons (alpha = 0.05/3) using G*power[90]. The results indicated that with a sample size of 95 we have enough power ($1$-$\beta = 0.8$) to detect the expected effect size ($r = 0.3$). The planned sample size therefore met this sample size requirement while also estimating stabilized effect sizes. Nine people failed attention checks built in the task, so their data was removed from any analysis, leaving a sample of 294 individuals (Mean age = 45.13, SD age = 16.31; 51.36% female, 47.96% male and 0.34% non-binary; 71.43% White, 12.24% Black or African American, 5.1% Asian). Detailed demographic information is provided in Supplementary Table S10.

**Experimental task.** The experimental task was identical to that used in Studies 2 and 3.

**Context-sensitivity index or CSI**[46]. CSI was identical to that used in Studies 2 and 3.

**Situational test of emotional understanding-brief or STEU-B**[41]. STEU-B was identical to that used in Study 3.

**Procedure.** The procedure of this study is identical to Study 3.

**Data preparation.** Following the same procedure from Study 2, participants who provided accidental, incorrect, or had missing ratings for more than 10% of their data were removed from analysis. Data from 19 participants was removed leaving a sample size of 275 participants. We also performed multivariate outlier detection using the Mahalanobis distance function from the *Routliers* package in R[92]. 7 subjects were identified as outliers and their data was removed resulting in 268 participants for final analysis.

## Study 5
**Overview.** Study 5 comprised of an experiment that was conducted twice with the same group of participants. This study had a within-subjects design with all participants responding to the same experiment.

**Participants.** 136 native English-speaking participants between the age of 18–60 years were recruited from the US using the online data collection platform Prolific. To determine sample size, we carried out power analysis for bivariate correlations controlling for multiple comparisons using G*Power[90]. The results indicated that with a sample of 102 we have enough power ($1$-$\beta = 0.8$) to detect a desirable effect size (ICC: $r = 0.75$). We oversampled for an additional 25% of data to account for dropouts, exclusions and attrition based on previous studies, resulting in a total sample of 136 individuals. Seventeen people dropped out as they did not complete Session 2 so their data was removed from any analysis, leaving a sample of 119 individuals (Mean age = 36.78, SD age = 11.29; 48.74% female, 47.06% male and 4.20% non-binary; 65.55% White, 8.40% Black or African American, 5.88% Asian). Detailed demographic information is provided in Supplementary Table S10.

**Experimental task.** The experimental task was identical to that used in Studies 2–4.

**Procedure.** In session 1, participants first read and signed an online consent form agreeing to participate in the Study. Then, participants read instructions for the experimental task, these were identical to those provided in Studies 2–3. This was followed by a question aimed to validate that participants read the instructions. Participants then performed the experimental task. Finally, participants filled out a brief demographic questionnaire and were thanked and compensated for their participation. The procedure for session 2 was identical to session 1. Session 2 was administered two weeks after session 1.

**Data preparation.** First, data was removed for participants who did not complete both sessions since computing test-retest reliability requires complete data for two time points. Next, following the same procedure from Studies 2–4, participants who provided accidental, incorrect, or had missing ratings for more than 10% of their data in either session 1 or 2 were removed from analysis. Data from 9 participants was removed leaving a sample size of 110 participants.

## Reporting summary
Further information on research design is available in the Nature Portfolio Reporting Summary linked to this article.

## Data availability

All data (deidentified) collected and analyzed for studies 1–5 and supplementary materials are provided in the OSF repository[95] (https://osf.io/7e6j5/). The archival data[24] used in this manuscript has been made publicly available by the original authors on OSF. We have included instructions in the readme file of our OSF repository on how to access that data and incorporate it into our analyses.

## Code availability

The code for all analyses that support the findings of this study are available at https://osf.io/7e6j5/.

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

## Acknowledgements

We thank Tuan Le Mau and Lisa Feldman Barrett for sharing their dataset and stimuli with us.

## Author contributions

S.G. and M.G. developed the study concept. S.G. and M.G. designed the initial archival analyses. S.G. and M.G. designed subsequent experiments. S.G. and M.G. planned remaining analysis and J.J.-E. and D.C.O. consulted on data analysis and supplementary data collection. S.G. collected and analyzed all experimental data. S.G. wrote the manuscript with contributions from M.G., J.J.-E., and D.C.O.

## Competing interests

The authors declare no competing interests.
