## [Peer Review File · Nature Communications]

Face and context integration in emotion inference is limited and variable across categories and individualsEditorial Note: Parts of this Peer Review File have been redacted as indicated to remove third-party material where no permission to publish could be obtained.

Reviewers' comments:

Reviewer #1 (Remarks to the Author):

The authors investigate a set of questions concerning the integration of situation information and facial expressions during the attribution of emotions. First, the study tests whether emotion attributions made by participants who have access to both information about the situation and a person's facial expression are best captured by a model that relies exclusively on facial expressions, by a model that relies exclusively on situations, or by a model that integrates expressions and situations. Among these three models, the model that relies exclusively on situations shows the best correlation with the data. The authors then show that this varies for different emotion categories and for different participants. The authors conclude that overall, people mostly rely on situation information alone for emotion attribution.

The study investigates important questions in the field of emotion attribution, and I commend the authors for their use of more realistic stimuli compared to previous studies. However, there are two major methodological concerns that lower my confidence in the conclusion reached in the study.

1) The authors propose to use a Bayesian model of cognition, and as such, the model should capture the cognitive processes at play during emotion inference. In the cue-integration equation, therefore, the probability $p(e|s)$ should stand for the probability with which a specific participant believes that the character in the stimuli experiences emotion e when exposed to situation s . Analogously, the probability $p(e|f)$ should stand for the probability with which a specific participant believes that the character in the stimuli experiences emotion e when they display facial expression f . However, participants were not asked about their uncertainty about emotion judgments. For example, in Study 1 "each participant responded (yes/no; followed by slightly, moderately, strongly, intensely) to the 13 different emotions." If I understand correctly, probabilities were computed calculating the frequencies of emotion ratings across multiple participants. Importantly, there is no guarantee that frequencies of emotion judgments across participants (i.e. the proportion of participants who reported that a character would experience a certain emotion in a certain situation) would correspond to the uncertainties of emotion representations within each individual participant. There are clear examples of cases in which the two do not match – this can occur for example if participants generate judgments using maximum-a-posteriori estimation. For this reason, a possible alternative explanation for the authors' results is that the Bayesian cue integration model is not correlating as well with the participants' judgments because the incorrect probabilities are being used in the model: instead of using distributions that capture each participant's uncertainty about the emotions, distributions that capture the frequency of emotion judgments across participants are used.

2) The Bayesian cue integration model in equation 1 relies on the assumption that the causal relationship between situations, emotions and expressions is captured by a Directed Acyclic Graph in which situations cause emotions, and emotions cause expressions. However, this is a simplifying assumption – in principle, expressions can be jointly caused by emotions and situations. For example, someone who feels shame might want to display that shame in some situations, but might want to hide it in other situations. If we consider this richer model in which expressions are not caused by emotions alone, but by emotions and situations jointly, equation 1 is incorrect. Indeed, in the derivation of equation 1 (see Ong and Zaki), $p(f|e,s)$ is treated as equal to $p(f|e)$ on grounds that the dependence between s and f is fully mediated by e (the simplifying assumption). (in Ong and Zaki, the letter o for outcome is used instead of s).

Taking this into account, an alternative explanation of the results reported in the present study is that the simplifying assumption that expressions depend exclusively on emotions (rather than jointly on emotions and situations) might have worked in the simpler scenarios in the study by Ong and Zaki (2015), but might not be adequate to capture inference in the more realistic scenarios considered here. The emotions categories for which the cue integration model performs worse than the situation only model might indeed be categories for which the same emotion might lead to different expressions

depending on the situation. In line with this possibility, shame is indeed one of them, as are pride and embarrassment – other kinds of emotions that individuals might want to display or conceal depending on the circumstances. In order to exclude this alternative explanation, the authors would need to test a model of cue integration that does not rely on the simplifying assumption $p(f|e,s) = p(f|e)$, and show that the situation only model outperforms even this more accurate cue-integration model.

Reviewer #2 (Remarks to the Author):

In the current manuscript, the authors present archival data and four new studies to investigate whether emotion inferences rely primarily on the face, the situation, or an integration of the two and in how far people differ in their reliance on these cues. The situation-only model fits the data best, yet there is variability between emotions and persons. On the one hand, I think the manuscript provides an extremely valuable contribution to the field. It tests its hypotheses using formal modeling in a more precise way, which is a very rare but desperately needed advancement over previous research. As another critical extension of previous research, the manuscript formally derives and tests individual differences in the ways people infer emotions. The evidence across sufficiently powered studies is strong and thought-provoking. Overall, I very much like this manuscript. On the other hand, I think the description of the methods and statistical tests could be improved. Potentially, I am simply misunderstanding a few things that could be ruled out with providing more details and rewriting a view sections. Finally, I have some minor points the authors could address. I elaborate on each of these points below.

Major point

1. I do not fully understand how all model parameters were estimated and to which ratings they were related. According to Figure 1a and the text, the Bayesian model is estimated with the formula. The situation-only model is estimated from people's ratings of situation-only stimuli (i.e., participants see only a situation description and rate whether and how much each emotion is present). And the face-only model is estimated from people's ratings of face-only stimuli (i.e., participants see only a face and rate whether and how much each emotion is present). For the critical tests of model fit, the authors correlate participant's ratings from the face+situation stimuli ratings (i.e., participants see a face and a situation and rate whether and how much each emotion is present) with the three model estimates above. Hence, to run such tests, each study ideally has all three conditions, namely a situation-only stimulus condition, a face-only stimulus condition, and a face+situation stimulus condition. The critical correlations are reported for all four studies (Table S1). However, according to the Methods section, Studies 2 to 4 had only a face+situation stimulus condition. Hence, if my description above is not mistaken, did the authors use all model parameter estimates only from Study 1 and correlate these with new ratings from face+situation stimuli in Studies 2 to 4? But doesn't this testing then require that the samples are comparable? Imagine Study 1 used a sample in which participants do not rely a lot on the face. Then, their face-only model estimates will also include lots of noise. Now in Study 4 potentially the sample relies a lot on the face. However, as their face+situation ratings are correlated with the noisy face-only model estimates from Study 1, the face-only model cannot show high model fit. I do not think this happened in the studies, yet, I also do not understand why the authors did not simply include all three conditions in all studies. Relatedly, if I am not mistaken, the authors estimated individual differences by correlating participants' ratings in each study with the model estimates from Study 1. This cross-study correlation may have the same problems as stated above. What I found confusing was that Figure 1b suggests that participants' data is correlated only with the face-only and the situation-only model estimates from Study 1. However, when reporting results, there is also a discussion in how far participant's ratings correspond to the Bayesian model. Is Figure 1b incomplete? Do I misunderstand things? In sum, I suspect that I am misunderstanding parts of the Methods and Results. I recommend four things. First, clarify in the main text where each model estimate came from and with which other ratings it was correlated (i.e., that model estimates came only from Study 1 and face+situation ratings

from all studies). Second, discuss the former point as a limitation (i.e., it requires that the samples are comparable, which cannot be ensured; could be addressed with always having all three conditions). Third, try showing that the samples are comparable (e.g., by showing that they have similar demographics or that their face+situation ratings converge). Fourth, clarify Figure 1b graphically and its description in the main text. Maybe some recommendations don't make sense or other changes make more sense. Overall, I recommend clarifying the Methods and Results a bit.

Minor points

2. line 106, typo; it should be "perceivers"

3. lines 739, 794, and 834; alpha is not the common symbol for power but for the alpha level of a significance test. Power is typically $1 - \beta$. I recommend changing to $1 - \beta$.

4. The adherence to open science principles is outstanding and comprehensive. I was only wondering about the names of the files on OSF. It is not entirely self-evident which file relates to which study. I recommend choosing better file names that link directly to study names (e.g., Analysis_Study1.Rmd) or adding a README file that clarifies where users would find which information.

5. Did the authors analyze the emotion-specific model fits only in Study 1? Why did they not analyze the fit in Studies 2 to 4?

6. I am curious to hear the authors' thoughts on three points. If they consider the points reasonable, they may decide to discuss them in the discussion. First, the models all assume that only one emotion is present. However, complex situations like the ones in the situations may trigger multiple emotions at the same time (e.g., Larsen & McGraw, 2014; Moeller et al., 2018; Trampe et al., 2015), in the current set for instance anger and contempt (two emotions that may frequently co-occur; e.g., Fischer & Giner-Sorolla, 2016). Can the models, in theory, be extended to capture such cases, for instance by estimating the belief that a combination of emotions is present given a situation, face, or combination of both? In the data, participants were also allowed to indicate that in a single stimulus multiple emotions were present. Second, the data misses a ground truth. So far, the evidence shows that people's ratings about face+situation stimuli converge more with ratings of situation-only stimuli than other ratings. Imagine I could ask the actors which emotion(s) of the 13 they actually displayed. Would also the situation-only model estimates converge most strongly with this ground truth? Put differently, do the results speak to the issue of emotion recognition accuracy or could the method be extended to investigate this question (I am tacitly neglecting the point whether determining such accuracy actually makes sense). Third, the research relies only on the face as expressive cue, which makes absolute sense given the dominance of the face in emotion inference research. Beyond the face, emotions can be expressed with the body, vocally, verbally, or in symbolic ways (reviewed in Lange et al., 2022). Would the situation-only model also outperform body, face+body, asf. models? How could the cue integration model be extended to include more cues next to face and situation? The authors describe such expressive cues as "contextual" (line 65), in line with other scholars (e.g., Aviezer et al., 2012). However, these are cues expressed by the person (as opposed to merely being present in the environment) and have been listed next to the face also in classic work (e.g., Levenson, 1994).

Aviezer, H., Trope, Y., & Todorov, A. (2012). Body cues, not facial expressions, discriminate between intense positive and negative emotions. *Science*, 338(6111), Article 6111.

<https://doi.org/10.1126/science.1224313>

Fischer, A., & Giner-Sorolla, R. (2016). Contempt: Derogating others while keeping calm. *Emotion Review*. <https://doi.org/10.1177/1754073915610439>

Lange, J., Heerdink, M. W., & Van Kleef, G. A. (2022). Reading emotions, reading people: Emotion perception and inferences drawn from perceived emotions. *Current Opinion in Psychology*, 43, 85–90.

<https://doi.org/10.1016/j.copsy.2021.06.008>

Larsen, J. T., & McGraw, A. P. (2014). The case for mixed emotions. *Social and Personality Psychology Compass*, 8(6), Article 6. <https://doi.org/10.1111/spc3.12108>

Levenson, R. W. (1994). Human emotions: A functional view. In P. Ekman & R. J. Davidson (Eds.), *The nature of emotions: Fundamental questions* (pp. 123–126). Oxford University Press.

Moeller, J., Ivcevic, Z., Brackett, M. A., & White, A. E. (2018). Mixed emotions: Network analyses of intra-individual co-occurrences within and across situations. *Emotion*, 18(8), Article 8.

<https://doi.org/10.1037/emo0000419>

Trampe, D., Quoidbach, J., & Taquet, M. (2015). Emotions in everyday life. *PLOS ONE*, 10(12), Article 12. <https://doi.org/10.1371/journal.pone.0145450>

Reviewer #3 (Remarks to the Author):

The authors present a study investigating the influence of contextual cues on emotional perception. I appreciate that the study was well-designed, and the motivation of the study was also easy to follow. Especially the analytic design is very innovative and clever in that it quantifies perception as a function of cue context (w/wo) and emotional categories (e.g., happiness), comparing the pattern similarity of models to anchor patterns. I believe the study results will provide an in-depth understanding of how we perceive emotions using various sources, such as facial features and situational awareness. Nonetheless, I would like a commentary and see what the authors think.

I think manipulating contextual cues and facial features would not be fair enough, as some facial expressions can be ambiguous. For example, previous studies showed how surprising or neutral facial expressions (although not used in this study) are easily influenced by the context due to their nature of ambiguity. So the rationale is that some facial expressions can depend more on the situational condition (or individual tendencies, such as depression). However, the situational cues in this study sound more likely to make it easier for individuals to recognize a highly consistent and accurate single emotional state as it is straightforward and language-based. Thus, it is possible that emotion perception when faces come with contextual cues, observers less utilize the facial features as the contextual category can be easier to understand. As a result, the responses were made simply by the language cue rather than facial features, which may require more effort. If there were less emotional or vague context cues, do observers still rely more on the context information? Or, if the contextual ratings existed, could they be considered as weight or parametric values in the model estimation? Please accept apologies if my understanding of the stimulus set was incorrect. However, details would be beneficial to future readers.

Previous studies suggested the age-related difference in utilizing contextual information in emotion perception. This is mainly due to the degeneration of the ability to integrate various information simultaneously. Similarly, there is also an age-related ambiguity tolerance of facial expressions. However, in the current study, the sample's age range seems very broad (up to 60), so it would be beneficial if the authors examined any age-related effects.

Minor comment: Did not the study with citation number 46 show the effect of an individual's emotional tendencies on how to use/be influenced by contextual information?

Reviewer #4 (Remarks to the Author):

In this paper the authors build and extend the computational lay theory of emotion of Ong et al (2015) to test emotional cue integration across situations and facial signals. The advancement of the present work is the use of more complex and rich scenarios as well as more diverse and naturalistic facial behavior. The authors compared each of the three model estimates (Face-only, Situation-only and Bayes cue-integration) to the empirical cue integration probabilities. They report Pearson correlations of empirical cue-integration probabilities with the model probabilities and conclude that the Situation-only model had the highest correlation, surpassing cue integration and face only models. Overall, the topic is important and the paper is well written. While I found the work interesting, there are several comments worth noting.

- 1) These results differ from those of Ong et al (2015) in which cue integration between outcomes and facial expressions performed better than face only and outcome only models (see Ong et al, 2015, Exp 3). However, the presently reported difference between the correlation of the situation only model ($r = 0.86$) and the cue integration model ($r = 0.84$) is very small (albeit significant). It seems that both models capture the empirical data quite well. In fact, it seems that in most tested cases, cue integration was at least as common as situation only judgments (Figure 3). This diverges significantly from the gist of the paper's conclusions.
- 2) Additionally, the question of cue integration would seem to depend on the clarity of each source independently. For this matter, it seems plausible that the present scenarios provide much more detailed information about the likely experienced emotions than the faces alone do. Thus, the current findings might be highly specific to the specific operationalization of "context", and may yield different results when the context is vaguer or the faces are clearer.
- 3) Furthermore, the authors describe the current study as using more complex and naturalistic stimuli, and I agree, certainly compared to computer generated faces and the narrow situations in previous work. However, the study is entirely based on acted portrayals of Hollywood actors which (from looking at some examples online) seem quite exaggerated and extreme (even if not stereotypical). It's unclear if these portrayals reflect human behavior in day-to-day life and if the broad conclusions of the paper are justified.
- 4) The paper also delves into individual differences in cue integration/situation only/face only responses. However, there is no report of stability across time for these patterns. Testing participants on the same task several days /weeks apart would be helpful in determining consistency which seems important if these individual differences are to be interpreted as meaningful.

Reviewer #1 (Remarks to the Author):

The authors investigate a set of questions concerning the integration of situation information and facial expressions during the attribution of emotions. First, the study tests whether emotion attributions made by participants who have access to both information about the situation and a person's facial expression are best captured by a model that relies exclusively on facial expressions, by a model that relies exclusively on situations, or by a model that integrates expressions and situations. Among these three models, the model that relies exclusively on situations shows the best correlation with the data. The authors then show that this varies for different emotion categories and for different participants. The authors conclude that overall, people mostly rely on situation information alone for emotion attribution.

The study investigates important questions in the field of emotion attribution, and I commend the authors for their use of more realistic stimuli compared to previous studies. However, there are two major methodological concerns that lower my confidence in the conclusion reached in the study.

1) The authors propose to use a Bayesian model of cognition, and as such, the model should capture the cognitive processes at play during emotion inference. In the cue-integration equation, therefore, the probability $p(e|s)$ should stand for the probability with which a specific participant believes that the character in the stimuli experiences emotion e when exposed to situations. Analogously, the probability $p(e|f)$ should stand for the probability with which a specific participant believes that the character in the stimuli experiences emotion e when they display facial expression f . However, participants were not asked about their uncertainty about emotion judgments. For example, in Study 1 “each participant responded (yes/no; followed by slightly, moderately, strongly, intensely) to the 13 different emotions.” If I understand correctly, probabilities were computed calculating the frequencies of emotion ratings across multiple participants. Importantly, there is no guarantee that frequencies of emotion judgments across participants (i.e. the proportion of participants who reported that a character would experience a certain emotion in a certain situation) would correspond to the uncertainties of emotion representations within each individual participant. There are clear examples of cases in which the two do not match – this can occur for example if participants generate judgments using maximum-a-posteriori estimation. For this reason, a possible alternative explanation for the authors' results is that the Bayesian cue integration model is not correlating as well with the participants' judgments because the incorrect probabilities are being used in the model: instead of using distributions that capture each participant's uncertainty about the emotions, distributions that capture the frequency of emotion judgments across participants are used.

We thank the reviewer for their comment. They are correct in their observation that we compute probabilities using the frequency of emotion ratings and thus are assuming that these frequency ratings would correspond to uncertainties.

To respond to the reviewer's concern, we collected, in a new sample, people's certainty ratings on all 13 emotion categories for a subset of the stimuli ($n = 44$) for face-only and situation-only

conditions. We computed the three model estimates using these certainty ratings (see Fig S4 below). We found that the pattern of model fits was similar when using our original frequency ratings and the newly collected certainty ratings. When we statistically compare model estimates, we find that the Bayesian model fit, and the situation-only model fit, are not statistically different ($t = 1.06, p = 0.29$), in contrast to the primary results we report in the manuscript. We do see that the RMSE is still lower for the situation-only model (RMSE = 0.042) compared to the Bayesian model (0.044), but the magnitude of difference is reduced compared to primary results. These findings suggest that there is no clear evidence for integration *over* simple reliance on context but that both models account for the empirical data. The conclusion that facial information is adding little to no value beyond situational cues alone suggests the more parsimonious account is that perceivers may be *generally* relying on situational information rather than integrating.

Below we present Figure 2 from the main text and Figure S4 from the supplementary materials to show the parallels in the findings across the two approaches.

Figure 2 | Overall comparison of the three models in Study 1. a, Overall Pearson model correlation for each model. Higher correlation values suggest better model fit. **b,** Overall root-mean-squared-error (RMSE) estimate for each model. Lower RMSE values suggest better model fit. The error bars represent bootstrapped 95% confidence intervals.

Figure S4 | Overall comparison of the three models using direct certainty ratings for a subset of stimuli ($n = 44$). **a**, Overall Pearson model correlation for each model – face-only ($r = 0.625$, 95% CI: 0.560, 0.686), Bayes cue-integration ($r = 0.867$, 95% CI: 0.839, 0.890), and Situation-only ($r = 0.854$, 95% CI: 0.823, 0.880). Higher correlation values suggest better model fit. The face-only model correlation is significantly different from Bayes cue-integration ($t = 16.52$, $p < 0.0001$) and Situation-only models ($t = 9.94$, $p < 0.0001$), but there is no statistically significant difference between Bayes cue-integration and Situation-only model ($t = 1.06$, $p = 0.29$). **b**, Overall root-mean-squared-error (RMSE) estimate for each model - face-only (0.063, 95% CI: 0.058, 0.069), Bayes cue-integration (0.044, 95% CI: 0.040, 0.049), and Situation-only (0.042, 95% CI: 0.039, 0.046). Lower RMSE values suggest better model fit. The error bars represent bootstrapped 95% confidence intervals.

We present the findings from certainty judgements in supplementary materials (Figure S4) and discuss this in the main manuscript on page 9:

“As a robustness test, we also computed model estimates using direct certainty judgements instead of intensity judgements and find that strong reliance on situational cues is not a limitation of our approach of computing frequency-based probability estimates (see Supplementary Figure S4). We do find that the Bayesian model fit, and the situation-only model fit, are not statistically different when using certainty judgments ($t = 1.06$, $p = 0.29$). This indicates that both models could plausibly account for how perceivers are inferring emotions, but the situation-only model includes fewer parameters and is thus a more parsimonious model candidate for emotion inferences. Perceivers likely rely more heavily on situational information, even when integrating.”

More broadly we have tempered claims throughout the manuscript regarding the fit of the Bayesian versus situation-only model. For details, please see our response to Reviewer 4, comment 1.

Further, we are now explicit in our methods section about the assumptions that frequency-based estimates are a valid way to derive probabilities given the literature on probability matching and Bayesian samplers. The text on page 6 now reads:

“We compute overall model probabilities by averaging emotion intensity ratings across participants. This assumes that the distribution of frequency ratings can be used to compute model probabilities, consistent with prior literature demonstrating people’s inferences reflect probability matching behavior¹⁻⁶ and that people act as Bayesian samplers⁷⁻⁹.”

Finally, we want to note that we continue to present probabilities computed using the frequency of emotion ratings and believe that relying on these probabilities is still a reasonable approach given that there is a large literature demonstrating people’s decisions and inferences often reflect probability-matching behavior¹⁻⁶. An example of probability matching is when people are asked to guess the outcome of flipping a biased coin (75% heads, 25% tails) over several trials. Experimentally, people choose their responses proportionally to their success probabilities (75% heads, 25% tails) even though the ‘optimal’ response would be to maximize and always (100% of the time) guess ‘heads’, which is the maximum-a-posteriori (MAP) estimate. Instead of picking the MAP estimate, people often respond with frequencies matching the underlying probability distribution in decision-making tasks, as well as when making other types of inferences. This probability-matching behavior can in fact be produced when people randomly sample a few hypotheses with frequency proportional to their probability instead of considering the full probability distribution. The idea that people act as these Bayesian samplers is a widely proposed explanation of how minds (and brains) perform Bayesian inference⁷⁻⁹. It posits that minds carry out rational probabilistic inference by using sample-based approximate inference given its processing limitations compared to the computationally expensive nature of exact Bayesian inference. We designed this project based on prior literature supporting the assumption that people are probability matching. While it is beyond the scope of the current project to weigh in on this larger debate on whether probabilistic inferences are based on frequency matching or MAP estimation, we believe that the close match of the intensity and certainty rating distributions suggests that our model estimates are robust to this concern.

2) The Bayesian cue integration model in equation 1 relies on the assumption that the causal relationship between situations, emotions and expressions is captured by a Directed Acyclic Graph in which situations cause emotions, and emotions cause expressions. However, this is a simplifying assumption – in principle, expressions can be jointly caused by emotions and situations. For example, someone who feels shame might want to display that shame in some situations, but might want to hide it in other situations. If we consider this richer model in which expressions are not caused by emotions alone, but by emotions and situations jointly, equation 1 is incorrect. Indeed, in the derivation of equation 1 (see Ong and Zaki), $p(f|e,s)$ is treated as equal to $p(f|e)$ on grounds that the dependence between s and f is fully mediated by e (the simplifying assumption). (in Ong and Zaki, the letter o for outcome is used instead of s). Taking this into account, an alternative explanation of the results reported in the present study is that the simplifying assumption that expressions depend exclusively on emotions (rather than jointly on emotions and situations) might have worked in the simpler scenarios in the study by Ong and Zaki (2015), but might not be

adequate to capture inference in the more realistic scenarios considered here. The emotions categories for which the cue integration model performs worse than the situation only model might indeed be categories for which the same emotion might lead to different expressions depending on the situation. In line with this possibility, shame is indeed one of them, as are pride and embarrassment – other kinds of emotions that individuals might want to display or conceal depending on the circumstances. In order to exclude this alternative explanation, the authors would need to test a model of cue integration that does not rely on the simplifying assumption $p(f|e,s) = p(f|e)$, and show that the situation only model outperforms even this more accurate cue-integration model.

The reviewer is correct that in the present paper, we are evaluating integration that assumes that facial cues are independent of the situational context in which they were generated. The present modeling approach was developed to test whether perceivers integrate in a manner consistent with the dominant accounts of emotion inference in the literature (for review, see Barrett et al., 2019). Specifically, both in theory and in practice, researchers assume that emotion categories and facial behaviors demonstrate stable, context-free links, such that there is an expression of anger (scowling) or an expression of fear (wide-eyed, gasping) that is invariant across the specific contexts in which these expressions might occur. Barrett and colleagues (2019) recently termed this approach the “common view” in the field and provide extensive documentation of the prevalence of this approach in basic science, clinical science, education, the tech sector, security, and in popular media and entertainment. Indeed, the field of “emotion recognition” is premised on this assumption, where decontextualized portrayals of emotion are presented to perceivers and “accuracy” in emotion inference is computed based on whether the perceiver judges them in line with a researcher stipulated category or norms generated by a separate group of subjects. The present study is designed to address the extent to which perceivers infer emotion, including integrate facial and situational information, in line with the *common view*. We have now revised and restructured parts of the introduction to better situate our study, which is a test of Bayesian cue integration based on this *common view* of emotion inference. These edits are on pages 2-3, below we provide snippets where new text was added (highlighted in blue):

“Despite the complexity of emotional events in the real world, research examining emotion inferences has disproportionately focused on how people process isolated canonical facial portrayals of emotion (or, facial expressions)¹⁰. **This approach, termed the *common view* is prevalent in basic science, clinical science, education, the tech sector, security, and in popular media and entertainment (for review see ref¹¹). Specifically, both in theory and in practice, researchers assume that emotion categories and facial behaviors demonstrate stable, context-free links. For example, this *common view* assumes that there is an expression of anger (i.e., scowling) that is invariant across the specific contexts in which these expressions might occur (e.g., in both a conflict with a romantic partner and when getting cut off in traffic). Indeed, the field of “emotion recognition” is premised on this assumption, where decontextualized canonical portrayals of emotion are presented to perceivers and “accuracy” is computed based on whether responses match the a priori category.”**

“Though prior studies demonstrate the crucial role of context in informing emotion inferences, most are limited because they have classically focused on dominance of

certain cues rather than integration across cues to form inferences (for a classic example on integration, see ref¹²). That is, earlier work was often aimed at establishing dominance of cues (facial or situational) by pitting them against one another as highly contrastive information (for example, see refs^{13,14}). **In this approach, there was frequent experimental use of pre-selected “clear” (meaning, eliciting high consensus in perceiver judgments) facial and situational cues (for critiques on clarity of stimuli, see ref¹⁵) which likely restricted the stimuli to more stereotypical displays (e.g., a wide-eyed, gasping expression for fear).”**

“Specifically, Ong and colleagues (2015) proposed a rational-observer model of how laypeople integrate multiple emotional cues, based upon other rational-observer models in human visual perception. This model assumes that observers hold a causal lay theory in which situational outcomes are assumed to cause emotions, which in turn cause facial expressions (described in more detail below). **This is, essentially, a Directed Acyclic Graph such that situations do not directly impact the form of the expression itself. This lay theory assumes that perceivers rely on mental representations of context-free emotional expressions, in line with the *common view* of emotion inference outlined earlier.** And, that observers make inference over this mental model when integrating emotional cues.”

The independence of facial expressions from context is an assumption in much of the literature, given that canonical forms of expressions are widely proposed rather than context-specific forms of facial expressions (for which we are unaware of a systematic descriptive base). As such, we agree that poor model fit may be attributable to the fact that the facial behaviors themselves are contextual and that a better model that captures integration in emotion inference would incorporate this. Indeed, the stimulus set shows remarkable within category variability in the underlying facial actions that the actors engage in¹⁶, allowing us to conduct a strong test of the premise that perceivers infer emotion in line with the *common view*.

We agree with the reviewer that an alternative cue integration model in which expressions are jointly caused by situations and emotions is an important direction, given that integration did not generally outperform the situation-only model (although for certain emotions and certain individuals, it did). We believe one contribution of this paper is demonstrating the limitations of the common view and the need for an alternative cue integration model. However, we believe that proposing a new contextualized cue integration model is beyond scope of the present work. One necessary advance to implement such a model is a systematic descriptive base of expressions of the same emotion category within different situations. While our stimulus set likely captures some within category variability, it is not optimally designed to test such a cue integration model. Thus, to propose such a new model, we would need to collect a new stimulus set with these properties, which seems more appropriate for a future project. We outline this limitation in scope and the potential utility of such an alternative model as a future direction in the discussion on page 18. See updated text below:

“In future work, it will be valuable to examine cue-integration using alternative theoretical assumptions that can be tested using Bayesian inference. Here we examine cue-integration stemming from the *lay intuition* that people expect emotion experiences

to be arising from situational outcomes and leading to facial portrayals¹⁷. This *lay intuition* suggests that expressive behavior directly flows from emotions (which are themselves situated) such that the expressive behavior may not be further constrained by knowledge of the situational context. But people may hold a lay theory that emotional expression is situated, suggesting that the cue integration model itself may have limitations. As an example, the non-verbal expression one might expect in an instance of anger at one's boss would be highly distinct from an expression of anger at a driver who cut you off in traffic. It would be fruitful to examine integration of facial and situational cues based on an alternative model that assumes facial cues are constrained by both emotion and the situation. Testing this model will necessitate building a naturalistic stimulus set that contains sufficient and systematic situational variation within each emotion category to model this complexity."

Reviewer #2 (Remarks to the Author):

In the current manuscript, the authors present archival data and four new studies to investigate whether emotion inferences rely primarily on the face, the situation, or an integration of the two and in how far people differ in their reliance on these cues. The situation-only model fits the data best, yet there is variability between emotions and persons. On the one hand, I think the manuscript provides an extremely valuable contribution to the field. It tests its hypotheses using formal modeling in a more precise way, which is a very rare but desperately needed advancement over previous research. As another critical extension of previous research, the manuscript formally derives and tests individual differences in the ways people infer emotions. The evidence across sufficiently powered studies is strong and thought-provoking. Overall, I very much like this manuscript. On the other hand, I think the description of the methods and statistical tests could be improved. Potentially, I am simply misunderstanding a few things that could be ruled out with providing more details and rewriting a few sections. Finally, I have some minor points the authors could address. I elaborate on each of these points below.

1. I do not fully understand how all model parameters were estimated and to which ratings they were related. According to Figure 1a and the text, the Bayesian model is estimated with the formula. The situation-only model is estimated from people's ratings of situation-only stimuli (i.e., participants see only a situation description and rate whether and how much each emotion is present). And the face-only model is estimated from people's ratings of face-only stimuli (i.e., participants see only a face and rate whether and how much each emotion is present). For the critical tests of model fit, the authors correlate participant's ratings from the face+situation stimuli ratings (i.e., participants see a face and a situation and rate whether and how much each emotion is present) with the three model estimates above. Hence, to run such tests, each study ideally has all three conditions, namely a situation-only stimulus condition, a face-only stimulus condition, and a face+situation stimulus condition. The critical correlations are reported for all four studies (Table S1). However, according to the Methods section, Studies 2 to 4 had only a face+situation stimulus condition. Hence, if my description above is not mistaken, did the authors use all model parameter estimates only from Study 1 and correlate these with new ratings from

face+situation stimuli in Studies 2 to 4? But doesn't this testing then require that the samples are comparable? Imagine Study 1 used a sample in which participants do not rely a lot on the face. Then, their face-only model estimates will also include lots of noise. Now in Study 4 potentially the sample relies a lot on the face. However, as their face+situation ratings are correlated with the noisy face-only model estimates from Study 1, the face-only model cannot show high model fit. I do not think this happened in the studies, yet, I also do not understand why the authors did not simply include all three conditions in all studies.

We agree with the reviewer that there is an assumption that the samples are comparable when using face-only and situation-only ratings from Study 1 to construct models in Studies 2 to 4. Given that the participants were all sampled online, are located in the US, and are primarily English speaking, we made the assumption the samples were comparable and did not collect the face-only and situation-only ratings again (since these were always collected between-subjects). While it is conventional in computational cognitive modeling work to collect such judgements of *priors* from independent samples^{18,19} we also agree that this assumption of comparability is import to evaluate.

To respond to this reviewer concern, we have now collected a new sample of situation-only and face-only ratings using the same sampling strategy used in Studies 2 to 4. We show in the supplementary materials (Figure S2 and S3) and below that these distributions are comparable. There is a statistically significant high correlation of archival ratings with newly collected empirical ratings for both situation-only ($r = 0.966, p < 0.0001$) and face-only ($r = 0.934, p < 0.0001$) conditions.

Figure S2 | Comparing situation-only rating distribution. Smoothed density distribution of situation-only ratings averaged across emotion for the common set of stimuli ($n = 44$). The rating distributions were obtained from archival data (part of Study 1) and an *empirical data* set that was additionally collected to validate that the samples collected for the current manuscript are comparable to the archival data. There is statistically significant high correlation between situation-only ratings for each stimulus from archival and empirical data ($r = 0.966, p < 0.0001$).

Figure S3 | Comparing face-only rating distribution. Smoothed density distribution of face-only ratings averaged across emotion for the common set of stimuli ($n = 44$). The rating distributions were obtained from archival data (part of Study 1) and an *empirical data* set that was additionally collected to validate that the samples collected for the current manuscript are comparable to the archival data. There is statistically significant high correlation between face-only ratings for each stimulus from archival and empirical data ($r = 0.934$, $p < 0.0001$).

Also, we did initially check that the distributions of joint-cue ratings were comparable between the original Study 1 dataset and that of Study 2 and 3 but did not include this detail in the materials. We have now added these visualizations to the supplementary materials to better support this decision (Figure S1, also see below).

Figure S1 | Comparing joint-cue rating distribution. Smoothed density distribution of joint-cue ratings averaged across emotion for the common set of stimuli ($n = 44$) used in the archival dataset (part of Study 1), Study 2, and Study 3. There is statistically significant high correlation

of joint-cue ratings for each stimulus from archival data with Study 2 ($r = 0.954, p < 0.0001$) and Study 3 ($r = 0.966, p < 0.0001$) data.

Finally, we also performed the Kolmogorov-Smirnov test, a more conservative test of distribution comparability, and report the results of this test in the Supplementary materials (page 2) as follows:

“We also performed a more conservative test of distribution comparability using the Kolmogorov-Smirnov test. The distribution of joint-cue ratings obtained from archival data were not significantly different from those obtained in Studies 2 ($D = 0.058, p = 0.297$) and 3 ($D = 0.052, p = 0.411$). The distribution of situation-only and face-only ratings obtained from archival data were significantly different from the empirical data (Situation-only: $D = 0.084, p = 0.036$, Face-only: $D = 0.087, p = 0.025$). Visual inspection of the distributions suggests that this statistical difference is driven by the zero inflation of situation-only and face-only ratings in the archival data compared to the empirical data. However, the distribution of ratings along the intensity scale (1 to 4) is largely similar. Additionally, the high positive correlation between these rating distributions indicate that the two samples are sufficiently comparable for constructing model estimates. Indeed, model estimates for Study 2 and Study 3 computed using the *empirical* face-only and situation-only ratings (Table S9) follow the same pattern of results as observed with archival ratings (Table S1).”

2. Relatedly, if I am not mistaken, the authors estimated individual differences by correlating participants' ratings in each study with the model estimates from Study 1. This cross-study correlation may have the same problems as stated above. What I found confusing was that Figure 1b suggests that participants' data is correlated only with the face-only and the situation-only model estimates from Study 1. However, when reporting results, there is also a discussion in how far participant's ratings correspond to the Bayesian model. Is Figure 1b incomplete? Do I misunderstand things?

Thank you for the helpful comment regarding the visualization of the method. We have updated the figure 1b and its caption to also capture that we are comparing participants' data to the Bayesian model estimates. The updated Figure 1 and its caption are in the manuscript and attached below.

[redacted]

Figure 1 | Schematic representation of the analysis approach. **a**, Overall performance of each model is estimated by comparing average judgements of emotions for each stimulus in the joint-cue condition with Bayes cue-integration, Situation-only and Face-only model estimates separately. **b**, Each individual's reliance on **cue-integration**, **situational** and **facial** cues was computed by comparing their judgement of emotions from the joint-cue condition with group level estimates of **Bayes cue-integration**, **Situation-only** and **Face-only** model **respectively**. **The group level estimates were** obtained from the archival dataset for a subset of stimuli. *Note:* We

are unable to provide example images of the facial portrayals because the images are under copyright.

3. In sum, I suspect that I am misunderstanding parts of the Methods and Results. I recommend four things. First, clarify in the main text where each model estimate came from and with which other ratings it was correlated (i.e., that model estimates came only from Study 1 and face+situation ratings from all studies).

We have added this as a clarification to the main text on page 6. It now reads:

“We compute Face-only, Situation-only and Bayes cue-integration model estimates in Study 1 and compare it with people’s judgements of emotions from viewing both facial and situational cues across Studies 1-5 (empirically collected for Studies 2-5). As a robustness check we collected an additional set of ratings for face-only and situation-only conditions to ensure that sampling variation did not significantly impact these model fits. We find that the distributions of ratings are highly comparable (see Supplementary Figures S1-S3), and the model fits are consistent using the additional rating set (see Supplementary Tables S1).”

4. Second, discuss the former point as a limitation (i.e., it requires that the samples are comparable, which cannot be ensured; could be addressed with always having all three conditions).

We have added the following text to page 18 of the manuscript discussing this as a limitation.

“Additionally, we use model estimates computed from Study 1 sample in Studies 2-4. This assumes that the inferences about emotion from these cues would be comparable across samples which could be a limitation (see robustness checks in Supplementary Figures S1-S3 and Supplementary Table S1).”

5. Third, try showing that the samples are comparable (e.g., by showing that they have similar demographics or that their face+situation ratings converge).

The demographic characteristics of samples collected in our studies are comparable to the Le Mau, et al., 2019 dataset as these samples were recruited using convenience sampling techniques from a pool of participants involved in research on online platforms. We have added the demographics from archival data to our supplementary materials (Table S11, also provided below) and added a description to the table note indicating how the sampling criteria are comparable across studies. Overall, the distribution of sample across categories of gender and race are largely comparable as observed from the percentage of sample reported in each category. We also have added to the supplement two additional sources of evidence that the data are comparable (see response to comments 1, 3 and, 4 above).

Table S11. Demographic details for participants from Studies 2-5 and archival data¹⁶.

Percentage of Participants in each sample	Age	Gender	Race and Ethnicity	Education
Study 2 (N = 142)	Mean = 33.84 SD = 11.69	M = 47.18 F = 50.70 Non-Binary = 2.11	American Indian or Alaskan Native = 0 Asian = 10.56 Black or African American = 7.75 Native Hawaiian or Other Pacific Islander = 0.7 White = 76.06 Hispanic = 0 Latinx = 0 Other/Mixed Race = 4.93	Less than a high school diploma = 1.41 High School Degree or Equivalent = 18.31 Some college = 23.94 Associate Degree = 7.75 Bachelor's Degree = 38.03 Postgraduate Degree = 10.56
Study 3 (N = 162)	Mean = 36.04 SD = 12.87	M = 34.57 F = 59.88 Non-Binary = 5.56	American Indian or Alaskan Native = 0 Asian = 5.56 Black or African American = 12.35 Native Hawaiian or Other Pacific Islander = 0.62 White = 71.6 Hispanic = 2.47 Latinx = 0 Other/Mixed Race = 7.41	Less than a high school diploma = 0 High School Degree or Equivalent = 12.35 Some college = 25.93 Associate Degree = 12.35 Bachelor's Degree = 39.51 Postgraduate Degree = 9.88
Study 4 (N = 294)	Mean = 45.13 SD = 16.31	M = 47.96 F = 51.36 Non-Binary = 0.34	American Indian or Alaskan Native = 0 Asian = 5.1 Black or African American = 12.24 Native Hawaiian or Other Pacific Islander = 0.62 White = 71.43 Hispanic = 3.74 Latinx = 0 Other/Mixed Race = 7.51	Less than a high school diploma = 0.34 High School Degree or Equivalent = 11.56 Some college = 23.13 Associate Degree = 11.56 Bachelor's Degree = 36.39 Postgraduate Degree = 16.33
Study 5 (N = 119)	Mean = 36.78 SD = 11.29	M = 47.06 F = 48.74 Non-Binary = 4.2	American Indian or Alaskan Native = 0 Asian = 5.88 Black or African American = 8.4	Less than a high school diploma = 0.84 High School Degree or Equivalent = 16.81 Some college = 26.05

			Native Hawaiian or Other Pacific Islander = 0	Associate Degree = 13.45
			White = 65.55	Bachelor's Degree = 31.09
			Hispanic = 3.36	Postgraduate Degree = 11.76
			Latinx = 0	
			Other/Mixed Race = 16.81	
Archival Situation-only (N = 839)	Median = 35	M = 43.27 F = 56.38 Other = 3.58	White = 80/45 Black or African American = 10.25 Asian = 4.65 Other = 4.65	NA
Archival Face-only (N = 842)	Median = 35	M = 41.92 F = 57.36 Other = 7.13	White = 79.10 Black or African American = 9.14 Asian = 6.53 Other = 5.23	NA
Archival Joint-cue (N = 845)	Median = 35	M = 42.6 F = 56.8 Other = 5.92	White = 79.17 Black or African American = 7.93 Asian = 7.69 Other = 5.21	NA

“Note: The sampling criteria across the archival data and empirical data collected in this project are highly similar. There is information on age, gender, race and ethnicity across the samples and the age of enrollment for all studies was restricted to 18-60 years. In the samples collected for this project, we ask an additional demographic question about education and have greater diversity in response options presented for race. Nevertheless, the distribution of sample across gender and race are largely comparable as observed from the percentage of sample reported in each category.”

6. Fourth, clarify Figure 1b graphically and its description in the main text. Maybe some recommendations don't make sense or other changes make more sense. Overall, I recommend clarifying the Methods and Results a bit.

See response to comment 2 above.

Minor points

7. line 106, typo; it should be “perceivers”

We have fixed this typo.

8. lines 739, 794, and 834; alpha is not the common symbol for power but for the alpha level of a significance test. Power is typically 1 – beta. I recommend changing to 1 – beta.

Thank you for pointing this mistake out, we have edited the symbol of power in the methods on pages 21-23.

9. The adherence to open science principles is outstanding and comprehensive. I was only wondering about the names of the files on OSF. It is not entirely self-evident which file relates to which study. I recommend choosing better file names that link directly to study names (e.g., Analysis_Study1.Rmd) or adding a README file that clarifies where users would find which information.

We have renamed the analysis files to reflect the respective study data that is being analyzed within. We have also included a Readme file that provides a description of the analysis files to better orient anyone viewing the files.

10. Did the authors analyze the emotion-specific model fits only in Study 1? Why did they not analyze the fit in Studies 2 to 4?

To investigate individual differences, we needed to collect within subject data. As a result, to make the study feasible, the size of the set of stimuli we could use to gather participant ratings dramatically decreased (from 604 to 44). We used a smaller, randomly drawn set of stimuli for the individual difference studies so that the stimulus set to be comparable across individuals (so that differences were not attributable to the underlying stimuli). Given this, we do not feel confident that these 44 stimuli adequately sample the necessary stimulus space needed to draw conclusions about emotion-specific inferences. We have added this as a future direction on page 19:

“Another future direction will be to examine whether there are emotion-specific individual differences, a question which was beyond the scope of the present study.”

I am curious to hear the authors’ thoughts on three points. If they consider the points reasonable, they may decide to discuss them in the discussion.

11. First, the models all assume that only one emotion is present. However, complex situations like the ones in the situations may trigger multiple emotions at the same time (e.g., Larsen & McGraw, 2014; Moeller et al., 2018; Trampe et al., 2015), in the current set for instance anger and contempt (two emotions that may frequently co-occur; e.g., Fischer & Giner-Sorolla, 2016). Can the models, in theory, be extended to capture such cases, for instance by estimating the belief that a combination of emotions is present given a situation, face, or combination of both? In the data, participants were also allowed to indicate that in a single stimulus multiple emotions were present.

The modeling approach does allow that a given stimulus can have a distribution across emotions, but we agree with the reviewer that modeling emotion inferences as if they are independent is a limitation of this approach. The necessity of such a multivariate emotion model would imply that the ratings of two emotions in combination yields some emergent meaning that is not captured by modeling these ratings independently. This is an interesting future direction to pursue. We have added a statement to the discussion on page 19:

“Finally, an interesting future direction to pursue is a modeling approach that accounts for the profile of emotion ratings for a given stimulus, instead of treating each emotion for each stimulus separately when evaluating model fit. For instance, an experience of ‘disgusted-anger’, or of moral-outrage²⁰, would not be treated the same for disgust and anger as separate emotions.”

12. Second, the data misses a ground truth. So far, the evidence shows that people’s ratings about face+situation stimuli converge more with ratings of situation-only stimuli than other ratings. Imagine I could ask the actors which emotion(s) of the 13 they actually displayed. Would also the situation-only model estimates converge most strongly with this ground truth? Put differently, do the results speak to the issue of emotion recognition accuracy or could the method be extended to investigate this question (I am tacitly neglecting the point whether determining such accuracy actually makes sense).

Both crowd consensus and target self-reports are widely used means of evaluating emotion inferences—the latter is what the reviewer has in mind, which can allow one to calculate the “accuracy” of people’s ratings with respect to target self-reports. We agree that “accuracy” is not possible to evaluate in our approach and that consensus can be very distinct from accuracy (e.g., in the context of trait impressions). We have added the following sentence to the discussion on page 17:

“Here, we focus on modeling individual differences in cue utilization and integration with a consensus-based approach. An alternative would be to use reports of the emotion experienced or expressed by the target (for discussion see²¹). It is possible that these two approaches may yield distinct results insofar that the shared cultural model captured by consensus judgments diverges from the expression-emotion links for a given target.”

13. Third, the research relies only on the face as expressive cue, which makes absolute sense given the dominance of the face in emotion inference research. Beyond the face, emotions can be expressed with the body, vocally, verbally, or in symbolic ways (reviewed in Lange et al., 2022). Would the situation-only model also outperform body, face+body, asf. models? How could the cue integration model be extended to include more cues next to face and situation? The authors describe such expressive cues as “contextual” (line 65), in line with other scholars (e.g., Aviezer et al., 2012). However, these are cues expressed by the person (as opposed to merely being present in the environment) and have been listed next to the face also in classic work (e.g., Levenson, 1994).

This is a great empirical question and extension of the current work. We have added this as an interesting future direction on pages 18-19:

“Another extension is to test the generalizability of the present findings across different types of cues, including when bodily cues serve as context. For example, based on prior work²², it is likely that a bodily-context model would at least similarly outperform the Face-only model. The current modeling approach can be extended to formalize lay beliefs about a range of cues in people’s understanding of emotions.”

Reviewer #3 (Remarks to the Author):

The authors present a study investigating the influence of contextual cues on emotional perception. I appreciate that the study was well-designed, and the motivation of the study was also easy to follow. Especially the analytic design is very innovative and clever in that it quantifies perception as a function of cue context (w/wo) and emotional categories (e.g., happiness), comparing the pattern similarity of models to anchor patterns. I believe the study results will provide an in-depth understanding of how we perceive emotions using various sources, such as facial features and situational awareness. Nonetheless, I would like a commentary and see what the authors think.

1. I think manipulating contextual cues and facial features would not be fair enough, as some facial expressions can be ambiguous. For example, previous studies showed how surprising or neutral facial expressions (although not used in this study) are easily influenced by the context due to their nature of ambiguity. So the rationale is that some facial expressions can depend more on the situational condition (or individual tendencies, such as depression). However, the situational cues in this study sound more likely to make it easier for individuals to recognize a highly consistent and accurate single emotional state as it is straightforward and language-based. Thus, it is possible that emotion perception when faces come with contextual cues, observers less utilize the facial features as the contextual category can be easier to understand. As a result, the responses were made simply by the language cue rather than facial features, which may require more effort. If there were less emotional or vague context cues, do observers still rely more on the context information? Or, if the contextual ratings existed, could they be considered as weight or parametric values in the model estimation? Please accept apologies if my understanding of the stimulus set was incorrect. However, details would be beneficial to future readers.

This is a very interesting question. We agree that language-based context descriptions may be more readily understood than inferences based on real-world contexts that need to be directly perceived. We do not have ratings of how ambiguous or elaborate the contexts are, but the perceiver-based consensus for situation-alone ratings can be used to indirectly quantify this. While there is variability across situations, the ratings tend to be more complex than face ratings overall. Further, situation ratings also have lower consensus on any given emotion compared to the face ratings. This can be observed, on average, from the greater variance of ratings for each emotion category in the situation-only condition compared to face-only condition (Figure S7 below). Further, the number of emotions endorsed i.e., when the median rating of an emotion was greater than 0 for a stimulus, is greater for situation-only condition compared to face-only condition, on average (Figure S5 below) as well as for a majority of stimuli (Figure S6 below).

Figure S5 | Average emotions endorsed for face-only and situation-only conditions. Distribution of emotions endorsed for each stimulus in the face-only and situation-only condition. Median rating greater than zero for any emotion on a given stimulus is quantified as an endorsed emotion for that stimulus.

Figure S6 | Difference in number of emotions endorsed by stimuli. Number of emotions endorsed for each stimulus in the face-only condition is subtracted from those endorsed in situation-only condition. Median rating greater than zero for any emotion on a given stimulus is quantified as an endorsed emotion for that stimulus. Values greater than 0 suggest a greater number of emotions were endorsed for that stimulus in the situation-only condition compared to the face-only condition.

Figure S7 | Average variance in ratings for each emotion across the face-only and situation-only conditions. Distribution of variance in ratings for each emotion category in the face-only and situation-only condition. For emotion category, on average there is greater variance of ratings in the situation-only condition compared to the face-only condition.

We added this information to the supplementary materials (Figures S5-S7) and also added a brief discussion to the main text. The text on page 18 now reads:

“Another limitation is that the situational cues here were all descriptions of social situations rather than dynamically unfolding situations as experienced in real-life. As such, the features of the situation that were highlighted for participants may not be as readily accessible in everyday life. Real world understanding of situations requires the ability to represent abstract features and attentional resources to prioritize relevant features (for example, see refs^{23,24}). Mitigating this concern is the relatively low consensus for situational ratings in our data (see Supplementary Figures S5-S7), which suggest that these cues have variable interpretation and are thus ambiguous.”

2. Previous studies suggested the age-related difference in utilizing contextual information in emotion perception. This is mainly due to the degeneration of the ability to integrate various information simultaneously. Similarly, there is also an age-related ambiguity tolerance of facial expressions. However, in the current study, the sample's age range seems very broad (up to 60), so it would be beneficial if the authors examined any age-related effects.

We agree that this is an important consideration and specifically sampled individuals in early to mid-life to avoid strong aging-related effects. However, we agree with the reviewer that by age 60 we could already have some of these aging effects in evidence. To address this point, we analyzed the data across Studies 2-4 to see whether age is related to cue-reliance or integration (it was not possible to examine in Study 1 as model fit was not computed at the participant level).

Overall, we do not find evidence for any age-related effects influencing the cue-reliance or cue-integration estimates. The results of this analysis are now presented and discussed in Supplementary materials under the header ‘Age-related effects’ as following:

Table S6. Results for Pearson correlation of cue-reliance and cue-integration with participants’ age across studies 2-4.

	Face-only model	Situation-only model	Bayes cue-integration model
Study 2	$r = -0.069 [-0.241, 0.106]$ $p = 0.439$	$r = 0.332 [0.167, 0.478]$ $p = 0.0001$	$r = 0.056 [-0.119, 0.229]$ $p = 0.532$
Study 3	$r = -0.001 [-0.162, 0.161]$ $p = 0.996$	$r = 0.167 [0.005, 0.320]$ $p = 0.043$	$r = 0.054 [-0.109, 0.214]$ $p = 0.514$
Study 4	$r = 0.111 [-0.009, 0.228]$ $p = 0.069$	$r = 0.264 [0.149, 0.373]$ $p < 0.0001$	$r = 0.094 [-0.026, 0.212]$ $p = 0.125$

“Note: Given previous studies demonstrating age-related effects on how context influences processing of facial expressions²⁵, we also conducted an exploratory analysis to examine such effects. Results indicate that participants’ age is not related to their reliance on facial cues or integration of facial and situational cues across studies 2-4 (see Table S9). However, there is a small but significantly positive correlation between age and reliance on situational cues. This suggests that as people grow older, they are more likely to rely on situational information, consistent with previous findings²⁵. Given the small but positive relationship between age and situation-reliance, we re-examined the association of situation-reliance with STEU-B controlling for age. Consistent with our initial results, we find positive association between situation reliance and people’s situated understanding of emotions after controlling for age (Study 3: $r = 0.29$, $p = 0.0004$; Study 4: $r = 0.31$, $p < 0.0001$). This suggests there is a robust association between people’s tendency to use situational information to infer other’s emotions and their ability to understand the different emotions that are experienced across various social situations.”

We have also added the following statement to the main text on page 11:

“For extended analysis on age related effects on model estimates see supplementary materials (Table S6 and the note below it).”

3. Minor comment: Did not the study with citation number 46 show the effect of an individual's emotional tendencies on how to use/be influenced by contextual information?

While this paper looked at contextual modulation, it did not look at integration/cue-utilization. We have revised how we describe this paper in the introduction to better capture the nature of the findings from this paper. The sentence on page 4 now reads:

“Even when multimodal cues are presented to participants (for example, Geneva Emotion Recognition Test²⁶), researchers have **only focused on the individual differences in the**

degree of contextual modulation of emotion inferences for a limited set of contrasting facial and contextual cues²⁷. It remains an open question whether there is variation in individuals' tendency to integrate complementary facial and situational cues when inferring emotions and the degree of cue reliance.”

Reviewer #4 (Remarks to the Author):

In this paper the authors build and extend the computational lay theory of emotion of Ong et al (2015) to test emotional cue integration across situations and facial signals. The advancement of the present work is the use of more complex and rich scenarios as well as more diverse and naturalistic facial behavior. The authors compared each of the three model estimates (Face-only, Situation-only and Bayes cue-integration) to the empirical cue integration probabilities. They report Pearson correlations of empirical cue-integration probabilities with the model probabilities and conclude that the Situation-only model had the highest correlation, surpassing cue integration and face only models. Overall, the topic is important and the paper is well written. While I found the work interesting, there are several comments worth noting.

1) These results differ from those of Ong et al (2015) in which cue integration between outcomes and facial expressions performed better than face only and outcome only models (see Ong et al, 2015, Exp 3). However, the presently reported difference between the correlation of the situation only model ($r = 0.86$) and the cue integration model ($r = 0.84$) is very small (albeit significant). It seems that both models capture the empirical data quite well. In fact, it seems that in most tested cases, cue integration was at least as common as situation only judgments (Figure 3). This diverges significantly from the gist of the paper's conclusions.

We thank the reviewer for pointing out this concern regarding the relative difference in model fit and agree that the magnitude of difference between the cue-integration and situation-only model in our manuscript differs from Ong and colleagues (2015). Our initial conclusion was that the situation-only model performed better than the cue-integration model. Overall, we do not have uniformly strong support for this claim. While we did observe statistical differences on average between these conditions, the correlation effect size was very similar. There are cases, i.e., for specific emotions (amusement, happiness) and for specific individuals (between 19% and 30% of the sample across studies), where Bayesian integration outperformed the situation-only model. Further, the cue-integration model only statistically underperformed relative to the situation model for a subset of emotions (*shame, pride, interest, embarrassment, and awe*) and the difference in effect size was small. Finally, when we computed model estimates using certainty judgements instead of intensity ratings, we find that the model fits are not statistically different (see our response to Reviewer 1 comment 1).

On the other hand, the second model performance indicator – RMSE—provided less favorable support for integration overall. The RMSE values of the two models (Bayes cue-integration: 0.071, bootstrapped 95% CI: 0.068, 0.073, Situation-only: 0.042, bootstrapped 95% CI: 0.041, 0.044) indicate that the average difference between the model predicted values and actual values is much smaller for the situation-only model compared to the cue-integration model. This

patterns of RMSE values was observed even when model estimates were computed using certainty judgements but the difference in values decreased (Bayes cue-integration: 0.044, 95% CI: 0.040, 0.049, Situation-only: 0.042, 95% CI: 0.039, 0.046). Further, emotion inferences drawn from situations are integrated into the cue-integration model (Eq 1) which could be driving the relatively good fit of the cue-integration model. Faces do not appear to be holding a lot of value in predicting inferences across datasets suggesting that they may not be adding more additional information. Moreover, the situation-only model includes fewer parameters and fits the empirical data to a similar degree when compared to the cue-integration model, suggesting it may be a more parsimonious model for emotion inferences.

Given the balance of evidence, we opted to make more conservative claims based on our results. As such, we have modified our conclusions and changed the language throughout to highlight that these two models performed similarly even when there were statistically significant differences. Below we provide snippets of texts from the paper where we made major changes (new text added is highlighted in blue).

Page 1:

“We applied a Bayesian cue-integration model to datasets that use a diverse and complex set of facial portrayals and social situations **to elicit emotion inferences**. Overall, our results suggest that when inferring other’s emotion states, **a model in which people rely only on knowledge of the situation provides a statistically significant but only slightly better fit for perceiver inferences than a Bayesian cue-integration model; both of which strongly outperform a model in which people rely on facial portrayals alone. These finding suggest that people could be integrating, but that inferences are just as well (and sometimes better) captured by knowledge of the situation alone.**”

Page 8:

“Pearson correlations of empirical cue-integration probabilities with the model probabilities indicate that although each of the three models had a significant and strong correlation with empirical judgements, the Situation-only model had the highest correlation ($r = 0.865$, $p < 0.0001$, 95% CI: 0.857, 0.873), followed by the Bayes cue-integration model ($r = 0.840$, $p < 0.0001$, 95% CI: 0.830, 0.849), and finally the Face-only model ($r = 0.660$, $p < 0.0001$, 95% CI: 0.644, 0.677) (Figure 2a). While there was a statistically significant difference in the Situation-only and Bayes cue-integration model correlations ($t = -7.19$, $p < 0.0001$; **based on a test of significance for a difference between two dependent correlations**), the difference in correlation values was relatively small suggesting similar model fit for integration and situation reliance. Compared to the Face-only model, we also observed a statistically significant difference between both the Situation-only ($t = 36.31$, $p < 0.0001$) and Bayes cue-integration ($t = 48.35$, $p < 0.0001$).”

Page 9:

“We replicated these overall findings in Studies 2-4, such that we observe that, on average, the Situation-only model had **a statistically higher correlation** with people’s judgements from both facial and situational cues (see Supplementary Table S2) **compared to the other two models. As a robustness test, we also computed model estimates using direct certainty judgements instead of intensity judgements and find that strong reliance on situational cues**

is not a limitation of our approach of computing frequency-based probability estimates (see Supplementary Figure S4). We do find that the Bayesian model fit, and the situation-only model fit, are not statistically different when using certainty judgments ($t = 1.06, p = 0.29$). This indicates that both models could plausibly account for how perceivers are inferring emotions, but **the situation-only model includes fewer parameters and is thus a more parsimonious model candidate for emotion inferences**. Perceivers likely rely more heavily on situational information, even when integrating.”

Page 11:

“In sum, in Study 1 we find that when using more **diverse and complex** situation descriptions and facial portrayals the Bayes cue-integration model **captures the empirical data well but** does not **consistently** outperform the Situation-only model for overall emotion inferences. This suggests that when people often have access to rich situational context, they tend to primarily rely on this information to infer emotions in others.”

Pages 15-16:

“Our data shows that overall inferences of other’s emotion states based on access to their facial and situational cues were **well aligned with both** inferences based on situational cues alone **and integration of cues**. On average, people’s emotion inferences based on situational cues alone had a significant **but only slightly** better fit with joint-cue empirical judgements compared to inferences based on the Bayesian cue-integration. **Further, compared to the situation-only and integration models, we found inferences based on facial cues alone were consistently inadequate in capturing emotion inferences made in presence of both faces and situations. This contradicts the common view notion of emotion inferences but importantly underscores the adequacy of rich situational knowledge in making nuanced inferences about other’s emotion states**. These findings suggest that people’s inferences about what someone else is feeling are **adequately** explained by their beliefs that situational context leads to emotion experience.”

2) Additionally, the question of cue integration would seem to depend on the clarity of each source independently. For this matter, it seems plausible that the present scenarios provide much more detailed information about the likely experienced emotions than the faces alone do. Thus, the current findings might be highly specific to the specific operationalization of “context” and may yield different results when the context is vaguer, or the faces are clearer.

Please see our response to reviewer 3, comment 1 above.

3) Furthermore, the authors describe the current study as using more complex and naturalistic stimuli, and I agree, certainly compared to computer generated faces and the narrow situations in previous work. However, the study is entirely based on acted portrayals of Hollywood actors which (from looking at some examples online) seem quite exaggerated and extreme (even if not stereotypical). It’s unclear if these portrayals reflect human behavior in day-to-day life and if the broad conclusions of the paper are justified.

We appreciate this reviewer comment, and we wholeheartedly agree that we cannot make the claim that these stimuli are representative of daily life experiences nor that they are necessarily “subtle”. Instead, what we do know is that the stimuli are a) variable, b) often non-canonical portrayals, c) generated by individuals with skill in this domain (actors).

To respond to the reviewer’s concern, we have reviewed the manuscript for any language that might imply the claim that the portrayals are naturalistic and have edited them to instead highlight that these are diverse and complex accordingly. Below we provide text from the manuscript where we have made such changes:

Page 3:

“This implies that the Bayesian integration model may not be as applicable to emotion inferences when the stimuli **have greater diversity and complexity**. Thus, we aim to test the robustness of the Bayesian integration account here.”

Page 8:

“In Study 1, we examined the robustness of the Bayesian model for cue-integration¹⁷ using a more **diverse and complex** set of stimuli.”

“We first examined whether the cue-integration model best captures people’s inference of emotions from these more **diverse and complex** facial and situational cues.”

Page 11:

“In sum, in Study 1 we find that when using more **diverse and complex** situation descriptions and facial portrayals the Bayes cue-integration model **captures the empirical data well but** does not **consistently** outperform the Situation-only model for overall emotion inferences.”

Page 16:

“The reliance on situation context may be particularly relevant **when thinking about** real-world emotion inferences. The cues used in our work have greater **diversity and complexity**. For example, portrayals of emotion in this dataset rarely conformed to a priori canonical expectations for how faces move¹⁶. For instance, it was rare that in situations that would be viewed as interest-inducing, actors portrayed that emotion by narrowing their eyelids and producing a controlled smile (the proposed canonical expression). **This could be because the actors were asked to enact the complexity of situations from the descriptions instead of providing them with an emotion category label or predetermined facial muscle movements to enact (as is common in the literature)**. Indeed, there was variability in how actors used their faces to portray the emotions. **However, these stimuli do not sufficiently capture the spontaneous naturalistic expressions that arise in everyday social situations**. These findings therefore underscore the importance of testing theories and computational models of emotions using diverse sets of more naturalistic stimuli. Such stimuli can help test people’s ability to infer emotions in a capacity that is more closely related to the real-world phenomenon.”

While we do agree that our stimuli are not naturalistic (and have made the above edits to temper this claim), we believe that these provide a more ecologically valid test of the computational model compared to those used in past work as also highlighted by the reviewer themselves. We clarify the reasoning for this in our introductions, the paragraph on page 3 now reads:

“We ask whether the Bayesian integration of facial and situational cues (based on the model from ref¹⁷) best accounts for perceiver’s emotion inferences compared to inferences based on single cues (face or context) alone when using stimuli with more variability and complexity. Prior modeling work was tested using a narrow set of situations: different outcomes (amount of money won) within a gambling game or outcomes (win or loss) from a game-show²⁸ or tennis match²⁹, sometimes along with computer-generated caricatured facial portrayals¹⁷. Thus, prior work leaves open the question of whether integration of facial and situational cues **based on the common-view assumptions** extends to more naturalistically varying stimuli that better reflect the diversity of experiences and expressive behaviors in everyday life. In this work, we test whether the existing model of cue-integration¹⁷ **based on the common view (heretofore referred to as integration)** accounts for emotion inferences in a data set of over 2500 perceiver judgments, where the situations are high in emotional complexity and the expressive behavior is highly variable yet high in perceived intensity¹⁶. Based on evidence that canonical expressions are rarely documented in everyday life²⁹ and previous research demonstrating that high-quality acted portrayals of emotion rarely involve canonical expressions²³, we contend that the present dataset provides a relatively more ecologically valid test of how perceivers integrate face and situations.”

Finally, we also discuss this important caveat on page 18:

“Our findings provide initial support for extending the examination of theories and computational models of emotions using **diverse and complex social scenarios and facial portrayals**, but more empirical work is required to address limitations of our approach. While the stimuli we employed here include more complexity and diversity than past work, they are not representative of the myriad of everyday experiences of emotion that occur. Further, the facial portrayals in the stimuli used are ultimately acted and do not capture actual spontaneous naturalistic expressions occurring in real-world contexts.”

4) The paper also delves into individual differences in cue integration/situation only/face only responses. However, there is no report of stability across time for these patterns. Testing participants on the same task several days /weeks apart would be helpful in determining consistency which seems important if these individual differences are to be interpreted as meaningful.

We thank the reviewer for this helpful suggestion and agree that having evidence of temporal stability would strengthen the interpretation of these results as individual differences. We have added a new test-retest sample (Study 5, $N = 110$) to determine if these individual differences are stable across a two-week time period. Overall, participants rated the same pairs ($n = 44$) of facial portrayals and situational descriptions as used in earlier studies (Studies 2-4) twice (session 1 and

session 2) over a two-week period. Analytical techniques, similar to that applied earlier (Studies 2-4), were used to compute individual's face-reliance, situation-reliance, and cue-integration estimates for both session one and two. Results for intra-class correlations of cue-reliance and cue-integration estimates across the two time points suggested, on average, a moderate test-retest reliability for reliance on situational cues alone ($ICC = 0.59, p < 0.0001, 95\% \text{ CI: } 0.22, 0.77$) and integration of cues ($ICC = 0.67, p < 0.0001, 95\% \text{ CI: } 0.51, 0.78$) but a weak reliability for reliance on facial cues alone ($ICC = 0.37, p < 0.0001, 95\% \text{ CI: } -0.17, 0.65$) (see Figure 7 below). In sum, we find that the situation-reliance and cue-integration estimates are moderately stable across time. But, face-reliance showed weak stability over time. Detailed description of results and methods are included on pages 14-15 and 23-24 respectively. We also discuss these findings in the paper on page 17:

“In addition to the variability across individuals, we also examined variability within individuals across time. Our findings suggest that over time people's reliance on situational cues and integration of cues are moderately stable tendencies. Such temporal stability suggests trait like tendencies indicating that variation in how individuals rely on situational cues and integration of cues are somewhat stable over time. In contrast, reliance on facial cues had low temporal stability. When examining the pattern of model fit estimates, we observe that facial cue reliance was generally higher at the second timepoint. This indicates that how people use facial cues to infer emotions may be contextual, such that familiarity drives up reliance on facial cues.”

Figure 7 | Test-retest reliability for cue-reliance and cue-integration estimates. Distribution of Fisher Z transformed model correlation values at two time points for **a**, Face-only model; **b**, Bayes cue-integration model and **c**, Situation-only model. The Pearson correlation values were transformed to z-scores using Fisher transformation before computing the intra-class correlations.

References

- 1 Wozny, D. R., Beierholm, U. R. & Shams, L. Probability matching as a computational strategy used in perception. *PLoS computational biology* **6**, e1000871 (2010).
- 2 Murray, R. F., Patel, K. & Yee, A. Posterior probability matching and human perceptual decision making. *PLoS computational biology* **11**, e1004342 (2015).
- 3 Gaissmaier, W. & Schooler, L. J. The smart potential behind probability matching. *Cognition* **109**, 416-422 (2008).

- 4 Unturbe, J. & Corominas, J. Probability matching involves rule-generating ability: a neuropsychological mechanism dealing with probabilities. *Neuropsychology* **21**, 621 (2007).
- 5 Thomas, E. A. & Legge, D. Probability matching as a basis for detection and recognition decisions. *Psychological Review* **77**, 65 (1970).
- 6 Kim, D. G. & Kim, H. C. Probability matching and strategic decision making. *Journal of Behavioral and Experimental Economics* **98**, 101850 (2022).
- 7 Vul, E. *Sampling in human cognition*, Massachusetts Institute of Technology, (2010).
- 8 Vul, E., Goodman, N., Griffiths, T. L. & Tenenbaum, J. B. One and done? Optimal decisions from very few samples. *Cognitive science* **38**, 599-637 (2014).
- 9 Goodman, N. D. & Tenenbaum, J. B. *Probabilistic Models of Cognition*. Vol. 2nd ed. (2016).
- 10 Gendron, M. & Barrett, L. F. Facing the past: A history of the face in psychological research on emotion perception. *The science of facial expression*, 15-36 (2017).
- 11 Barrett, L. F., Adolphs, R., Marsella, S., Martinez, A. M. & Pollak, S. D. Emotional expressions reconsidered: Challenges to inferring emotion from human facial movements. *Psychological science in the public interest* **20**, 1-68 (2019).
- 12 Goodenough, F. L. & Tinker, M. A. The relative potency of facial expression and verbal description of stimulus in the judgment of emotion. *Journal of Comparative Psychology* **12**, 365 (1931).
- 13 Wallbott, H. G. In and out of context: Influences of facial expression and context information on emotion attributions. *British Journal of Social Psychology* **27**, 357-369 (1988).
- 14 Carroll, J. M. & Russell, J. A. Do facial expressions signal specific emotions? Judging emotion from the face in context. *Journal of personality and social psychology* **70**, 205 (1996).
- 15 Ekman, P., Friesen, W. & Ellsworth, P. in *P. Ekman, WV Freisen/PK Ellsworth*. NY 269 (1972).
- 16 Le Mau, T. *et al.* Professional actors demonstrate variability, not stereotypical expressions, when portraying emotional states in photographs. *Nature communications* **12**, 1-13 (2021).
- 17 Ong, D. C., Zaki, J. & Goodman, N. D. Affective cognition: Exploring lay theories of emotion. *Cognition* **143**, 141-162 (2015).
- 18 Tessler, M. H. & Goodman, N. D. The language of generalization. *Psychological review* **126**, 395 (2019).
- 19 Kao, J. T., Wu, J. Y., Bergen, L. & Goodman, N. D. Nonliteral understanding of number words. *Proceedings of the National Academy of Sciences* **111**, 12002-12007 (2014).
- 20 Salerno, J. M. & Peter-Hagene, L. C. The interactive effect of anger and disgust on moral outrage and judgments. *Psychological science* **24**, 2069-2078 (2013).
- 21 Gendron, M. & Barrett, L. F. Emotion perception as conceptual synchrony. *Emotion Review* **10**, 101-110 (2018).
- 22 Aviezer, H., Trope, Y. & Todorov, A. Body cues, not facial expressions, discriminate between intense positive and negative emotions. *Science* **338**, 1225-1229 (2012).

- 23 Horstmann, K. T. & Ziegler, M. Situational perception: Its theoretical foundation, assessment, and links to personality. *The Wiley handbook of personality assessment*, 31-43 (2016).
- 24 Rauthmann, J. F. *et al.* The Situational Eight DIAMONDS: a taxonomy of major dimensions of situation characteristics. *Journal of Personality and Social Psychology* **107**, 677 (2014).
- 25 Ngo, N. & Isaacowitz, D. M. Use of context in emotion perception: The role of top-down control, cue type, and perceiver's age. *Emotion* **15**, 292 (2015).
- 26 Schlegel, K., Grandjean, D. & Scherer, K. R. Introducing the Geneva emotion recognition test: an example of Rasch-based test development. *Psychological assessment* **26**, 666 (2014).
- 27 Lee, T.-H., Choi, J.-S. & Cho, Y. S. Context modulation of facial emotion perception differed by individual difference. *PLOS one* **7**, e32987 (2012).
- 28 Houlihan, S. D., Ong, D., Cusimano, M. & Saxe, R. in *Proceedings of the Annual Meeting of the Cognitive Science Society*.
- 29 Anzellotti, S., Houlihan, S. D., Liburd Jr, S. & Saxe, R. Leveraging facial expressions and contextual information to investigate opaque representations of emotions. *Emotion* **21**, 96 (2021).

REVIEWER COMMENTS

Reviewer #1 (Remarks to the Author):

The authors have addressed many of my concerns, including additional data that substantially strengthened the paper. I believe they have also addressed the important concerns raised by Reviewer 4. In my view, a few additional clarifications are needed.

1.

The authors note:

"We computed the three model estimates using these certainty ratings (see Fig S4 below). We found that the pattern of model fits was similar when using our original frequency ratings and the newly collected certainty ratings."

and

"Finally, we want to note that we continue to present probabilities computed using the frequency of emotion ratings and believe that relying on these probabilities is still a reasonable approach given that there is a large literature demonstrating people's decisions and inferences often reflect probability-matching behavior¹⁻⁶."

The collection and analysis of certainty ratings addresses my concern regarding the use of frequencies. I thank the authors for including these data in the revised version of the manuscript, I believe that these data substantially strengthen the article, because results showing that people's decisions reflect probability-matching behavior in some perceptual tasks might not apply to these emotion inferences, especially considering that recent work reporting examples of cases in which people's decisions do not reflect probability-matching behavior in emotion judgments (Anzellotti, Houlihan, Liburd and Saxe 2020).

2. The authors note:

"When we statistically compare model estimates, we find that the Bayesian model fit, and the situation-only model fit, are not statistically different ($t = 1.06$, $p = 0.29$), in contrast to the primary results we report in the manuscript. We do see that the RMSE is still lower for the situation-only model (RMSE = 0.042) compared to the Bayesian model (0.044), but the magnitude of difference is reduced compared to primary results. These findings suggest that there is no clear evidence for integration over simple reliance on context but that both models account for the empirical data. The conclusion that facial information is adding little to no value beyond situational cues alone suggests the more parsimonious account is that perceivers may be generally relying on situational information rather than integrating."

I am not concerned about the lack of significance in the difference between the RMSE values, and I agree with the authors' conclusion that in the cases tested in this study facial information does not provide additional information beyond the situations.

However, this does not necessarily imply that perceivers generally rely on situational information rather than integrating cues. It might only indicate that faces provide a subset of the information that can be inferred from situations, therefore when complete information about both the face and the situation is available, faces might add little to no value. However, in realistic situations perceivers often do not have complete knowledge of the situation. In these cases, facial information might provide essential information. For example, imagine seeing a child receiving a gift and looking upset. If the perceiver knows that the child's sister was just given ten gifts, the upset facial expression might add

little value for emotion inference. However, if the perceiver does not have that information, the facial expression might well provide value in addition of the situation. Indeed, the facial expression might lead perceivers to make an inference about latent properties of the situation that might have caused an emotion that can account for the observed expression.

Cases in which perceivers have incomplete knowledge of the situation seem common, therefore, in order to study the integration of information about situations and facial expressions, it would be important to specifically test such cases. Alternatively, in order to support the conclusion that participants rely exclusively on situational information "generally", one would need compelling evidence showing that, in contrast with what one might expect, perceivers usually have complete knowledge of all the aspects of a situation that are relevant for emotion inferences.

Without such evidence, one can only conclude that the information participants were able to extract from faces was a subset of the information that they were able to extract from situations, and therefore when rich information about both faces and situations was available, a model that used both did not outperform a model using situations only at accounting for behavioral judgments. However, it is unknown whether information was extracted from both cues, or whether information was only extracted from situations, ignoring the faces. An appeal to parsimony is not sufficient to decide between these alternative possibilities: they are empirically testable hypotheses. To test them, one would need to design a stimulus set such that facial expressions provide unique information in addition to the available knowledge of the situation, and examine whether in such cases the perceivers' judgments are better captured by the model that combines both cues or by the model that relies only on situations. I understand that the authors' position might be that such stimuli are rare and unnatural, and therefore that such an experiment might not provide a valid understanding of naturalistic emotion inferences. However, in this case, the position that such stimuli are so rare would need to be supported by empirical evidence.

Reading the edits to the Discussion session made in response to Reviewer 4's comments – which largely align with my earlier comments in stating that previously the gist of the conclusions did not seem to align with the empirical data – I have seen that the authors have adjusted the language in the manuscript to recognize that the data do not rule out the integration model:

"This indicates that both models could plausibly account for how perceivers are inferring emotions, but the situation-only model includes fewer parameters and is thus a more parsimonious model candidate for emotion inferences. Perceivers likely rely more heavily on situational information, even when integrating."

This addresses my concern about the manuscript (while I still do not fully agree with the authors' response quoted above).

3. On lines 60-62, the "common view" is defined as an approach focusing disproportionately on how people process facial expressions in isolation:

"research examining emotion inferences has disproportionately focused on how people process isolated canonical facial portrayals of emotion (or, facial expressions)¹³. This approach, termed the common view"

On lines 100-105, assuming a DAG that lacks a direct connection from situations to expressions is stated to be "in line with the common view":

"This model assumes that observers hold a causal lay theory in which situational outcomes are assumed to cause emotions, which in turn cause facial expressions (described in more detail below). This is, essentially, a Directed Acyclic Graph such that situations do not directly impact the form of the

expression itself. This lay theory assumes that perceivers rely on mental representations of context-free emotional expressions, in line with the common view of emotion inference outlined earlier.”

However, assuming a DAG with structure SITUATION → EMOTION → EXPRESSION should lead to equation (1), which does not necessarily focus disproportionately on expressions. By contrast, using equation (1), both situations and expressions are used to infer emotions. Therefore, if I understand correctly, the assumption that there isn't a direct connection from situations to expressions in the DAG is not in line with the “common view”. It is another, different assumption.

To provide an example of a section of the text that would need to be adjusted to clarify this distinction, the manuscript states:

“prior work leaves open the question of whether integration of facial and situational cues based on the common-view assumptions”

However, this sentence seems to refer to the assumptions about the DAG, not to the use of isolated canonical facial portrayals of emotions as stated in the initial definition of the “common-view”.

It would be important to differentiate between these two assumptions clearly throughout the text.

Reviewer #2 (Remarks to the Author):

I had the opportunity to review the revision of the manuscript. The authors addressed all my points to my full satisfaction. As far as I can evaluate it, I also think that the authors addressed all points raised by Reviewer 4. I also want to repeat that I think this manuscript is of very high quality, substantially extends previous research, and paves the way for more advanced, formal work on emotion inferences.

I only have a very minor point. The authors mention that the time points in Study 5 were two weeks apart when briefly describing the methods on p.5. However, they do not mention the two weeks in the method section. For completeness of the method section, I recommend also mentioning the two weeks there.

REVIEWER COMMENTS

Reviewer #1 (Remarks to the Author):

The authors have addressed many of my concerns, including additional data that substantially strengthened the paper. I believe they have also addressed the important concerns raised by Reviewer 4. In my view, a few additional clarifications are needed.

1. The authors note:

“We computed the three model estimates using these certainty ratings (see Fig S4 below). We found that the pattern of model fits was similar when using our original frequency ratings and the newly collected certainty ratings.”

and

“Finally, we want to note that we continue to present probabilities computed using the frequency of emotion ratings and believe that relying on these probabilities is still a reasonable approach given that there is a large literature demonstrating people’s decisions and inferences often reflect probability-matching behavior¹⁻⁶.”

The collection and analysis of certainty ratings addresses my concern regarding the use of frequencies. I thank the authors for including these data in the revised version of the manuscript, I believe that these data substantially strengthen the article, because results showing that people’s decisions reflect probability-matching behavior in some perceptual tasks might not apply to these emotion inferences, especially considering that recent work reporting examples of cases in which people’s decisions do not reflect probability-matching behavior in emotion judgments (Anzellotti, Houlihan, Liburd and Saxe 2020).

We thank the reviewer for their comment and agree that the certainty judgement data substantially strengthens our manuscript. To indicate the value of such a comparison, we have now added text to the main manuscript highlighting the importance of understanding underlying decision processes for accurately modeling emotion judgements. The text on page 9 now reads:

“As a robustness test, we also computed model estimates using direct certainty judgements instead of intensity judgements and find that strong reliance on situational cues is not a limitation of our approach of computing frequency-based probability estimates (see Supplementary Figure S4). We find that the Bayesian model fit, and the situation-only model fit, are not statistically significantly different when using certainty judgments ($t(570) = 1.06, p = 0.29$) based on a two-tailed test of significance for difference between two correlations. We therefore continue to present probabilities computed using the intensity judgements, but would like to note that decisions reflecting probability matching behavior may not always apply to emotion inferences³⁶. To accurately model emotion judgements, it is important to understand the underlying decision processes that may reflect different representations across different perceptual tasks³⁶.”

2. The authors note:

“When we statistically compare model estimates, we find that the Bayesian model fit, and the situation-only model fit, are not statistically different ($t = 1.06$, $p = 0.29$), in contrast to the primary results we report in the manuscript. We do see that the RMSE is still lower for the situation-only model (RMSE = 0.042) compared to the Bayesian model (0.044), but the magnitude of difference is reduced compared to primary results. These findings suggest that there is no clear evidence for integration over simple reliance on context but that both models account for the empirical data. The conclusion that facial information is adding little to no value beyond situational cues alone suggests the more parsimonious account is that perceivers may be generally relying on situational information rather than integrating.”

I am not concerned about the lack of significance in the difference between the RMSE values, and I agree with the authors’ conclusion that in the cases tested in this study facial information does not provide additional information beyond the situations.

However, this does not necessarily imply that perceivers generally rely on situational information rather than integrating cues. It might only indicate that faces provide a subset of the information that can be inferred from situations, therefore when complete information about both the face and the situation is available, faces might add little to no value. However, in realistic situations perceivers often do not have complete knowledge of the situation. In these cases, facial information might provide essential information. For example, imagine seeing a child receiving a gift and looking upset. If the perceiver knows that the child’s sister was just given ten gifts, the upset facial expression might add little value for emotion inference. However, if the perceiver does not have that information, the facial expression might well provide value in addition of the situation. Indeed, the facial expression might lead perceivers to make an inference about latent properties of the situation that might have caused an emotion that can account for the observed expression.

Cases in which perceivers have incomplete knowledge of the situation seem common, therefore, in order to study the integration of information about situations and facial expressions, it would be important to specifically test such cases. Alternatively, in order to support the conclusion that participants rely exclusively on situational information “generally”, one would need compelling evidence showing that, in contrast with what one might expect, perceivers usually have complete knowledge of all the aspects of a situation that are relevant for emotion inferences.

Without such evidence, one can only conclude that the information participants were able to extract from faces was a subset of the information that they were able to extract from situations, and therefore when rich information about both faces and situations was available, a model that used both did not outperform a model using situations only at accounting for behavioral judgments. However, it is unknown whether information was extracted from both cues, or whether information was only extracted from situations, ignoring the faces. An appeal to parsimony is not sufficient to decide between these

alternative possibilities: they are empirically testable hypotheses. To test them, one would need to design a stimulus set such that facial expressions provide unique information in addition to the available knowledge of the situation, and examine whether in such cases the perceivers' judgments are better captured by the model that combines both cues or by the model that relies only on situations. I understand that the authors' position might be that such stimuli are rare and unnatural, and therefore that such an experiment might not provide a valid understanding of naturalistic emotion inferences. However, in this case, the position that such stimuli are so rare would need to be supported by empirical evidence.

Reading the edits to the Discussion session made in response to Reviewer 4's comments – which largely align with my earlier comments in stating that previously the gist of the conclusions did not seem to align with the empirical data – I have seen that the authors have adjusted the language in the manuscript to recognize that the data do not rule out the integration model:

“This indicates that both models could plausibly account for how perceivers are inferring emotions, but the situation-only model includes fewer parameters and is thus a more parsimonious model candidate for emotion inferences. Perceivers likely rely more heavily on situational information, even when integrating.”

This addresses my concern about the manuscript (while I still do not fully agree with the authors' response quoted above).

We thank the reviewer for their comment and address the important points raised that question the scope and application of our findings. We agree that to support the conclusions that people rely *exclusively* on situational information *in general*, in comparison to facial information, we would need either one of the following pieces of evidence. First, we would need evidence suggesting that regardless of the degree of ambiguity of situational context, when context is present, people's emotion inferences are based on that context, overriding their inferences drawn from corresponding facial cues. Or, second, we would need evidence demonstrating that the vast majority of human experiences consist of complete knowledge of the context.

To address this, we have now checked our manuscript and supplementary materials for any such claims of overgeneralized reliance on situational information. We have ensured that the interpretation of our finding - reliance on situational information for inferring emotions, does not overgeneralize. We have modified the following paragraph in our main manuscript where clarification was required:

Page 9:

“We replicated these overall findings in Studies 2-4, such that we observe that, on average, the Situation-only model had a statistically higher correlation with people's judgements from both facial and situational cues (see Supplementary Table S2) compared to the other two models. This indicates that perceivers rely more heavily on situational information, even when given the opportunity to integrate, possibly as facial information adds little to no value beyond the knowledge of the context.”

The note quoted above was from our previous response to the reviewers and we would like to clarify that as well:

“When we statistically compare model estimates, we find that the Bayesian model fit, and the situation-only model fit, are not statistically different ($t = 1.06$, $p = 0.29$), in contrast to the primary results we report in the manuscript. We do see that the RMSE is still lower for the situation-only model (RMSE = 0.042) compared to the Bayesian model (0.044), but the magnitude of difference is reduced compared to primary results. The conclusion that facial information is adding little to no value beyond situational cues alone suggests that even when integrating people rely more heavily on context information when both these cues provide sufficient information to infer other’s emotions.”

Further, the reviewer suggested some important ideas that would be interesting to examine to further build our understanding of reliance on and integration of cues in emotion inferences. Examining the boundary conditions of our findings by manipulating the ambiguity and amount of knowledge presented in the context is an important future direction and we have now included this in our discussion section on page 19 (text provided below). Further, we would like to note that while we agree with the reviewer that people may not always have complete knowledge of the context, adult human experiences are rarely devoid of contextual information and prior knowledge. Both in the example presented by the reviewer and the one we now include in our manuscript (text provided below); people have some contextual information available. These could be in the form of expectations that arise from knowledge of the event structure (e.g., birthday celebration, poker game), knowledge of the people involved (e.g., the other poker player rarely expresses their emotions), and their typical behavior (e.g., people deceive in poker games). This knowledge based on context may shape the meaning drawn from visual cues and the certainty of people’s emotion inferences. Quantifying the role of such sources of contextual information is certainly a high priority empirical question open for exploration. We also add this to our discussion section on page 18 (text provided below). Also, in our discussion, we previously noted that examining more ecologically valid sources of cues (e.g., visual scenes instead of textual descriptions) would be important to fully understand the generalizability of our findings as they pertain to reliance on knowledge of the situation (see below excerpt from page 18). We believe this is also relevant to addressing the reviewer’s comment.

Pag 19:

“Another extension is to test the generalizability of the present findings across different types of cues, including when bodily cues serve as context. For example, based on prior work⁶⁶, it is likely that a bodily-context model would at least similarly outperform the Face-only model. The current modeling approach can be extended to formalize lay beliefs about a range of cues in people’s understanding of emotions. The generalizability of our findings can also be tested by examining the extent to which reliance on single cues applies to varying degrees of contextual and facial information. An open question is whether people rely more on visual cues (e.g., faces or body language) in the presence of incomplete contextual information. For instance, when playing a game of poker with strangers at a casino, you do not have access to your opponent’s cards. Such lack of context information

may lead you to rely on visual cues to understand what the opponent might be feeling and use that to then reason about the context (e.g., do they have good or bad cards).”

Page 18:

“Another limitation is that the situational cues here were all descriptions of social situations rather than dynamically unfolding situations as experienced in real-life. As such, the features of the situation that were highlighted for participants may not be as readily accessible in everyday life. Real world understanding of situations requires the ability to represent abstract features and attentional resources to prioritize relevant features (for example, see refs^{82,83}). Mitigating this concern is the relatively low consensus for situational ratings in our data (see Supplementary Figures S5-S7), which suggest that these cues have variable interpretation and are thus ambiguous. **Further, even when individuals have incomplete or uncertain knowledge about a particular situation, it is likely that aspects of the perceived physical environment still form the basis for social predictions. As a result, it is unlikely that many real-world instances of emotion perception are ever truly decontextualized but testing cue-reliance and integration in different contexts with varying degrees of informational uncertainty is a question open for exploration.**”

3. On lines 60-62, the “common view” is defined as an approach focusing disproportionately on how people process facial expressions in isolation: “research examining emotion inferences has disproportionately focused on how people process isolated canonical facial portrayals of emotion (or, facial expressions)13. This approach, termed the common view”

On lines 100-105, assuming a DAG that lacks a direct connection from situations to expressions is stated to be “in line with the common view”:

“This model assumes that observers hold a causal lay theory in which situational outcomes are assumed to cause emotions, which in turn cause facial expressions (described in more detail below). This is, essentially, a Directed Acyclic Graph such that situations do not directly impact the form of the expression itself. This lay theory assumes that perceivers rely on mental representations of context-free emotional expressions, in line with the common view of emotion inference outlined earlier.”

However, assuming a DAG with structure SITUATION → EMOTION → EXPRESSION should lead to equation (1), which does not necessarily focus disproportionately on expressions. By contrast, using equation (1), both situations and expressions are used to infer emotions. Therefore, if I understand correctly, the assumption that there isn’t a direct connection from situations to expressions in the DAG is not in line with the “common view”. It is another, different assumption.

To provide an example of a section of the text that would need to be adjusted to clarify this distinction, the manuscript states:

“prior work leaves open the question of whether integration of facial and situational cues based on the common-view assumptions”

However, this sentence seems to refer to the assumptions about the DAG, not to the use of isolated canonical facial portrayals of emotions as stated in the initial definition of the “common-view”.

It would be important to differentiate between these two assumptions clearly throughout the text.

We thank the reviewer for bringing this to our attention and clarify the manuscript to specify what aspect of the *common view* we believe is relevant for the DAG (edited manuscript text is provided below), in alignment with the reviewer’s comment. One of the assumptions of the *common view* of emotions is that there are stable, context-free links between isolated facial expressions and emotion categories. This suggests that emotions can be inferred *accurately* (mostly measured as group consensus rather than true accuracy) from isolated facial expressions alone. This assumption of a relationship between expressions and emotions is present in the DAG as reflected by the direct link between emotions to expressions. However, as the reviewer pointed out, the DAG based on lay theory is not limited to this assumption and does not prioritize expressions over context as it also includes a link between context and emotions. The cue-integration model also includes the representation of situated emotions, unlike the *common view*, suggesting that knowledge of the context also provides information about emotion states.

The *common view* also includes the assumption of 1:1 mappings between specific expressions and emotions. This is not an underlying assumption of the cue-integration model based on DAG. While expressions are assumed to arise from emotion states, the DAG allows for links between multiple expressions and a given emotion state.

To summarize, we believe that the lay theory informing the DAG is in line with one aspect of the common view – a link between decontextualized facial expressions and emotion states but, is not entirely based on the *common view* nor does it imply no role of context in emotion inference.

We have edited the text in our manuscript in the following places to clarify this point:

Page 2:

“Despite the complexity of emotional events in the real world, research examining emotion inferences has disproportionately focused on how people process isolated canonical facial portrayals of emotion (or, facial expressions)¹³. This approach, termed the *common view* is prevalent in basic science, clinical science, education, the tech sector, security, and in popular media and entertainment (for review see ref¹⁴). Specifically, both in theory and in practice, researchers **often** assume that emotion categories and facial behaviors demonstrate stable, context-free links.”

Pages 2-3:

“The present research is designed to address integration of cues in emotion inferences using a computational modeling approach **that compares a cue-integration model to simpler models depending on facial information or situational information alone.**”

Page 3:

“Specifically, Ong and colleagues (2015) proposed a rational-observer model of how laypeople integrate multiple emotional cues, based upon other rational-observer models in human visual perception. This model assumes that observers hold a causal lay theory in which situational outcomes cause emotions, which in turn cause facial expressions (described in more detail below). This model extends beyond assumptions of the *common view*. The *common view* would suggest that facial expressions provide diagnostic information about the emotion that caused them. When situational outcomes are available, these should only provide convergent, overlapping information about the underlying emotion to that derived from the face. As such, an integration model and model based only on the face should capture emotion judgments to the same degree. In contrast, if the cue-integration model captures judgments to a greater degree than the face-only model, this implies that facial cues provide incomplete information about emotion, and that the situational context is providing additional information that perceivers utilize to infer emotion. Given that this lay theory is, essentially, a Directed Acyclic Graph, it should be noted that situations do not directly impact the form of the expression itself: instead, situations affect expressions only through the emotion itself. The cue-integration model (Equation 1 below) derived from this DAG thus assumes perceivers draw on context-independent mental representations of emotional expressions in the face, in line with an aspect of the *common view* outlined earlier³¹.”

Page 3:

“Thus, prior work leaves open the question of whether integration of facial and situational cues, based on the *Bayesian DAG model*, extends to more naturalistically varying stimuli that better reflect the diversity of experiences and expressive behaviors in everyday life.”

Page 16:

“Further, compared to the situation-only and integration models, we found inferences based on facial cues alone were consistently inadequate in capturing emotion inferences made in presence of both faces and situations. One interpretation is that, in our study context, the information that facial expressions contain are a (small) subset of the information in the situation context, such that face-only inference is poor, and cue-integrated inferences do not differ much from situation-only inferences. This contradicts one aspect of the *common view*: that facial behaviors are tightly linked to an emotion state, such that access to facial cues should provide diagnostic information about the underlying emotion. These findings suggest instead that people’s inferences about what someone else is feeling are adequately explained by their beliefs that situational context leads to emotion experience. These results additionally point to the need for a richer lay theory of emotions: perceivers may in fact consider expressions to be situated (i.e., in the causal model, adding a causal link between situations and expressions).”

Reviewer #2 (Remarks to the Author):

I had the opportunity to review the revision of the manuscript. The authors addressed all my points to my full satisfaction. As far as I can evaluate it, I also think that the authors addressed all points raised by Reviewer 4. I also want to repeat that I think this manuscript is of very high quality, substantially extends previous research, and paves the way for more advanced, formal work on emotion inferences.

We thank the reviewer for their consideration of our work.

I only have a very minor point. The authors mention that the time points in Study 5 were two weeks apart when briefly describing the methods on p.5. However, they do not mention the two weeks in the method section. For completeness of the method section, I recommend also mentioning the two weeks there.

Thank you so much for pointing this out. We have added this information under ‘Procedure’ subsection in the methods section. The text on page 24 now reads:

“Procedure. In session 1, participants first read and signed an online consent form agreeing to participate in the Study. Then, participants read instructions for the experimental task, these were identical to those provided in Studies 2-3. This was followed by a question aimed to validate that participants read the instructions. Participants then performed the experimental task. Finally, participants filled out a brief demographic questionnaire and were thanked and compensated for their participation. The procedure for session 2 was identical to session 1. Session 2 was administered two weeks after session 1.”

REVIEWERS' COMMENTS

Reviewer #1 (Remarks to the Author):

I thank the authors for the edits to their manuscript. The responses address my questions in full.